# AdaManip: Adaptive Articulated Object Manipulation Environments and Policy Learning

**Yuanfei Wang**[*1], **Xiaojie Zhang**[*2], **Ruihai Wu**[*1], **Yu Li**[2], **Yan Shen**[1], **Mingdong Wu**[1],
**Zhaofeng He**[2], **Yizhou Wang**[1 3 4 5], **Hao Dong**[1]

[1] Center on Frontiers of Computing Studies, School of Computer Science, Peking University
[2] Beijing University of Posts and Telecommunications
[3] Inst. for Artificial Intelligence, Peking University
[4] Nat'l Eng. Research Center of Visual Technology, Peking University
[5] State Key Laboratory of General Artificial Intelligence, Peking University

## Abstract

Articulated object manipulation is a critical capability for robots to perform various tasks in real-world scenarios. Composed of multiple parts connected by joints, articulated objects are endowed with diverse functional mechanisms through complex relative motions. For example, a safe consists of a door, a handle, and a lock, where the door can only be opened when the latch is unlocked. The internal structure, such as the state of a lock or joint angle constraints, cannot be directly observed from visual observation. Consequently, successful manipulation of these objects requires adaptive adjustment based on trial and error rather than a one-time visual inference. However, previous datasets and simulation environments for articulated objects have primarily focused on simple manipulation mechanisms where the complete manipulation process can be inferred from the object's appearance. To enhance the diversity and complexity of adaptive manipulation mechanisms, we build a novel articulated object manipulation environment and equip it with 9 categories of objects. Based on the environment and objects, we further propose an adaptive demonstration collection and 3D visual diffusion-based imitation learning pipeline that learns the adaptive manipulation policy. The effectiveness of our designs and proposed method is validated through both simulation and real-world experiments. Our project page is available at: `https://adamanip.github.io`

## 1 Introduction

Among the various categories of objects in our daily life, articulated objects are highly significant as they are common in our surroundings (such as cabinets, doors, and laptops) and their components are complex, featuring rich and diverse geometries, semantics, articulations, and functions. Therefore, learning articulated object representation (Du et al., 2023; Wei et al., 2022; Heppert et al., 2023; Lei et al., 2024) and manipulation (Xu et al., 2022; Wu et al., 2022) are essential while challenging for future robots in home-assistant tasks.

Among articulated object manipulation tasks, door manipulation (Urakami et al., 2019) is first and most thoroughly studied, as doors are most common and useful in our daily lives. Afterwards, with the release of diverse articulated object manipulation datasets and environments (Mo et al., 2019; Xiang et al., 2020; Liu et al., 2022; Geng et al., 2023b), various manipulation tasks (like opening, sliding, rotating, and further language-guided manipulation) on many categories of articulated objects (such as pots, lamps, and cabinets) have been studied.

While previous covered various aspects of articulated object manipulation, one of the most essential features of articulated objects, the mechanisms of different parts and articulations for accomplishing the final manipulation goal, has yet to be explored. For example, a safe can be directly opened by pulling the door in previous environments, while in the real world, the robot may have to first turn the key to unlock the latch, and then pull open the door. While UniDoorManip (Li et al., 2024b) proposes an environment with the corresponding dataset that can simulate the mechanisms of doors,

---

* indicates equal contribution

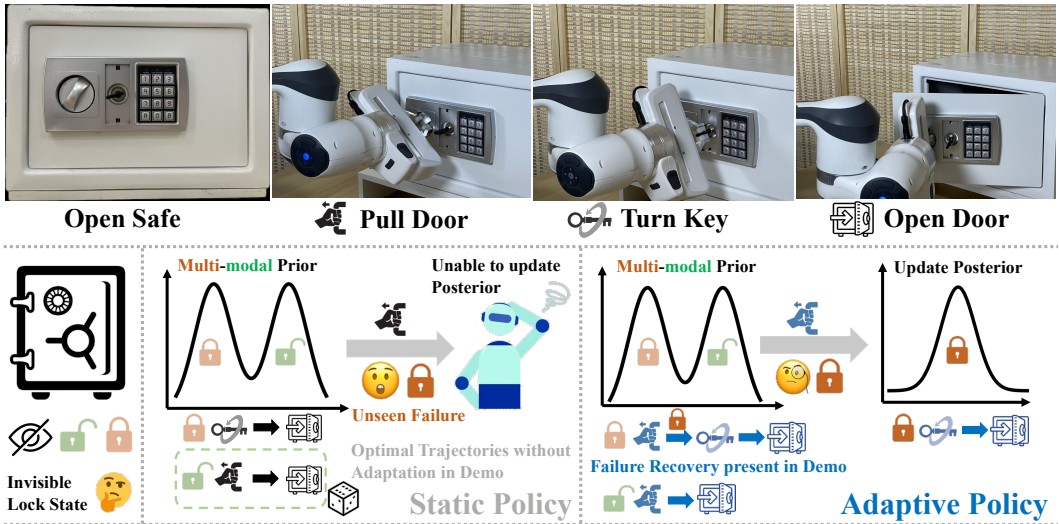

Figure 1: Example comparison between **Static** and **Adaptive Policies**. The safe can be directly opened if unlocked; otherwise, the key must be turned to unlock the latch before opening the door. However, it is impossible to figure out the lock state from pure visual observations. **Static Policy:** The demonstrations for training the static policy are optimal trajectories under full observation, including both locked and unlocked states. Consequently, the learned policy is a bimodal distribution based on visual observation alone. If the robot samples the "unlocked trajectory" and fails to open the locked door, it will be out of distribution. **Adaptive Policy:** The demonstrations for training the adaptive policy include recovery from the failed door opening. Therefore, the policy learns to first pull the door to check the lock state and updates the policy distribution accordingly based on the feedback.

the mechanisms of various types of articulated objects could be much more diverse and complicated. Therefore, we build an environment that can simulate the above-described complex mechanisms of articulated object manipulation, equipping this environment with 9 categories of different objects covering 5 types of adaptive mechanisms (details described in Section 3).

The different mechanisms of articulated objects call for two core capabilities of the policy: (1) multi-modal action proposal and (2) adaptive manipulation from history actions. For an observed object, the manipulation policy may contain multiple modes, including different manipulation trajectories. For example, when observing the safe with its door closed (Bottom-Left in Figure 1), to achieve the goal of opening the door, the policy could be either directly pulling the door (when the door is unlocked) or turning the key and then pulling the door (when the door is locked), and the method should be able to model these modalities from the same visual observation.

Furthermore, to identify and execute the accurate action from the multi-modal manipulation action candidates, it is necessary to adapt the manipulation policy based on previous actions and their corresponding outcomes. For example, when pulling the door results in no movement, the Adaptive policy should adapt from proposing multi-modal actions (either pulling the door or turning the key) to single-modal action (turning the key) (Bottom-Right in Figure 1).

To support multi-modal action proposals, we take advantage of the designs of diffusion policy (Chi et al., 2023) and its following studies (Ze et al., 2024; Yan et al., 2024; Tie et al., 2025), which have demonstrated modeling multi-modal distributions from only a few successful demonstrations. To empower the policy with adaptive manipulation abilities, while previous studies only collect optimal success trajectories (without any failures during the manipulation) for training, we introduce trajectories including failure actions and the recovery and adaptation actions from failures for training, as the failure actions help in revealing the accurate mechanisms and manipulation policy. Figure 1 showcases the superiority of our proposed adaptive policy learning method. The pure visual observation could not tell whether the door was locked or not. The static policy that only takes the passive one-frame visual input will randomly propose one of the multi-modal trajectory candidates. On the contrary, the adaptive policy will first try pulling the door, and then adapt the policy distribution from multi-modal to single-model accordingly, as its training data include the

failure of a direct pulling trial on the locked safe door, and then adaptively turning the key to open the safe after the failure successfully.

Based on our proposed novel environment, we have conducted extensive experiments on 9 categories of 277 different objects, covering 5 types of mechanisms, showcasing the necessity of the proposed environment and dataset, and the effectiveness of the proposed policy learning framework in efficiently and intelligently adapting the manipulation.

In summary, our contributions include:

- We study the novel problem of adaptively manipulating articulated objects with diverse mechanisms and build an environment with various categories of objects and mechanisms.
- We propose a novel framework that learns the adaptive manipulation policy for various mechanisms from diverse demonstrations.
- Extensive experiments have demonstrated the significance of our proposed environment, and the effectiveness of the proposed adaptive policy learning framework.

## 2 RELATED WORK

### 2.1 ARTICULATED OBJECT ENVIRONMENTS AND DATASETS

To facilitate the study of representation and manipulation of diverse and complex articulated objects, DoorGym (Urakami et al., 2019), a door manipulation environment is first introduced with diverse doors. UniDoorManip (Li et al., 2024b) further empowers door environments with different mechanisms. PartNet-Mobility dataset first introduces multiple categories of articulated objects from PartNet (Mo et al., 2019; Chang et al., 2015), integrated with the sapien environment (Xiang et al., 2020; Mu et al., 2021; Gu et al., 2023) to support various articulated object manipulation tasks. Further, GAPartNet (Geng et al., 2023b) provides fine-grained part annotations, AKB-48 (Liu et al., 2022) provides real-world articulated object models, and Arnold (Gong et al., 2023) provides the environment for language-guided manipulation.

### 2.2 ARTICULATED OBJECT MANIPULATION

There have been a series of studies studying articulated object manipulation. Where2Act (Mo et al., 2021) first studies the point-level affordance for short-term manipulation, with affordance-based (Wu et al., 2022), flow-based (Eisner et al., 2022; Zhang et al., 2023), part-based (Geng et al., 2023a) and rl-based (Geng et al., 2023c) methods study the long-horizon manipulation. Environment-Aware Affordance (Wu et al., 2023a; Li et al., 2024a) further studies the manipulation with environment constraints. Where2Explore (Ning et al., 2023) and AdaAfford (Wang et al., 2022) study converting passive visual priors to manipulation posteriors using the few-shot interactions, which tackle the problem of exploring novel articulated object categories with novel geometries and parts, and manipulation on ambiguous kinematics and dynamics. Besides, coarse-to-fine method (Ling et al., 2024) studies the sim2real framework for real-world manipulation, and language-guided methods (Xu et al., 2024; Gong et al., 2023) explore the manipulation with language guidance. While these works mainly investigated the manipulation with simple mechanisms (such as directly opening a door or safe), in our work, we further study the policy for manipulating articulated objects with diverse and complex mechanisms, with a novel proposed environment supporting such objects.

## 3 ADAPTIVE MANIPULATION ENVIRONMENT

Previous datasets and simulation environments for articulated objects often lack diversity and realistic manipulation mechanisms (Urakami et al., 2019; Li et al., 2024b; Geng et al., 2023b; Xiang et al., 2020; Geng et al., 2023a). To address this issue, we developed a new environment to better explore complex mechanisms in articulated object manipulation and learn adaptive manipulation policies. Based on IsaacGym (Makoviychuk et al., 2021), this environment simulates these mechanisms and includes 9 categories of objects (Section 3.1) with 5 types of adaptive mechanisms (Section 3.2).

### 3.1 ARTICULATED OBJECT DATASET

In recent years, several works have proposed datasets for articulated object manipulation. PartNet-Mobility (Xiang et al., 2020) and AKB-48 (Liu et al., 2022) offer diverse datasets for articulated

Table 1: Statistics of our adaptive articulated object dataset, including 9 categories of 277 different instances. **CM. PC.** respectively denote Coffee Maker and Pressure Cooker.

| Category | Bottle | Pen | CM. | Window | Door | Lamp | Microwave | Safe | PC. |
|---|---|---|---|---|---|---|---|---|---|
| **Instance** | 32 | 36 | 18 | 30 | 57 | 25 | 37 | 36 | 6 |

Table 2: Adaptive manipulation mechanism comparison between our environment and others.

| Environment | Lock | ±Clockwise | Rotate&Slide | Push/Rotate | Switch Contact |
|---|---|---|---|---|---|
| GAPartNet | ✗ | ✗ | ✗ | ✗ | ✗ |
| PartManip | ✓ | ✗ | ✗ | ✗ | ✗ |
| DoorGym | ✓ | ✗ | ✗ | ✗ | ✗ |
| UniDoorManip | ✓ | ✓ | ✗ | ✗ | ✗ |
| **Ours** | ✓ | ✓ | ✓ | ✓ | ✓ |

objects but focus on cross-category geometry diversity, neglecting the mechanisms of different parts and articulations needed to achieve the final manipulation goal. For instance, PartNet-Mobility includes the safe category, but the door can be directly opened without rotating the knob to unlock the latch. GAPartNet (Geng et al., 2023b) provides fine-grained part annotations but still fails to model complex manipulation mechanisms. DoorGym (Urakami et al., 2019) claims a large-scale, scalable dataset specifically for door manipulation, considering the latch mechanism of doors. UniDoorManip (Li et al., 2024b) enriches the diversity of door geometry by composing instances. However, both DoorGym and UniDoorManip are limited to door manipulation and do not cover more diverse and long-term mechanisms.

To investigate real-world articulated object manipulation, we introduce a new dataset that encompasses more realistic adaptive manipulation mechanisms. Our dataset includes 9 categories of 277 objects: **Bottle**, **Pen**, **Coffee Maker**, **Window**, **Pressure Cooker**, **Lamp**, **Door**, **Safe**, and **Microwave**. Table 1 provides detailed statistics of our dataset, and Figure 2 visualizes instances of each category. The object assets in our dataset are handcrafted from materials primarily obtained from 3D Warehouse (Trimble). More details can be found in Appendix B.

## 3.2 ADAPTIVE MANIPULATION MECHANISM

Most existing articulated object environments focus primarily on geometric diversity across different categories of objects. While these objects may contain multiple parts, manipulating one part typically does not impact other parts' state or joint limits, resulting in simplified manipulation mechanisms. Common actions in these environments include pushing or pulling a part, such as opening a drawer or pressing a button, which can be deduced purely from visual observation. However, real-world manipulation often depends on internal joint states that are not visible externally, necessitating adaptive manipulation policies based on feedback.

To better simulate real-world articulated object manipulation as well as corresponding mechanisms, we have identified five adaptive mechanisms that enhance the fidelity of our environment:

**Lock Mechanism**: Common in everyday objects like doors or safes, the lock mechanism requires an initial action such as rotating a key or knob or pressing a button to unlock the object before it can be opened. This mechanism tracks the key part's joint state during manipulation and updates the lock state accordingly. If the lock state transitions to "unlock", the door joint limit is lifted to allow opening; otherwise, the door remains locked. Since the lock state cannot be inferred visually, the robot must interact with the object to determine the lock state and adapt its policy accordingly.

**Random Rotation Direction**: When rotating a knob, cap, or handle, the direction (*i.e.*, clockwise or counterclockwise) is determined by the internal revolute joint limit. Our environment randomly assigns the rotation direction upon initialization, preventing visual inference of the direction. The robot must attempt one direction and switch if unsuccessful.

**Rotate & Slide Mechanism**: This mechanism requires a part to be rotated to a specific angle before it can be lifted or pulled out, such as lifting the lid of a pressure cooker. The required rotation angle is not visually discernible, necessitating the robot to rotate the part incrementally and attempt to slide it

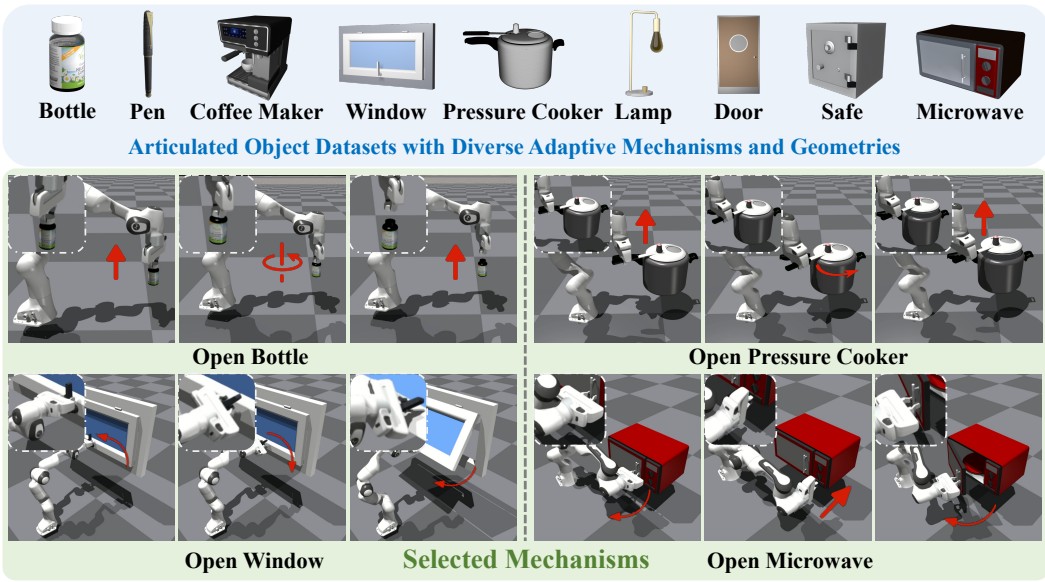

Figure 2: Adaptive manipulation dataset and environments. Bottle and Pressure Cooker feature the **Rotate & Slide** mechanism, requiring continued rotation after a failed lift. The window includes the **Lock** and **Random Rotation Direction** mechanisms, necessitating exploration of the correct rotation direction to unlock the latch. Microwave incorporates the **Lock** and **Switch Contact** mechanisms, where the robot must first pull the handle to check the lock state and press the button if locked.

to determine if the correct angle is reached. We randomize the revolute joint limit and initially set the prismatic joint limit to zero, lifting it once the correct angle is achieved.

**Push/Rotate Mechanism**: Due to the similar appearance of buttons and knobs, it is unclear whether the part should be pushed or rotated. For example, a lamp might be turned on by pushing a button in some instances and rotating a knob in others. Our environment includes both revolute and prismatic joints for the same part in the URDF file and randomly determines whether the part should be pushed or rotated, adjusting the joint limit accordingly.

**Switch Contact Mechanism**: Based on the lock mechanism, this requires the robot to manipulate different key parts, such as a handle and a knob, if they are separate. For instance, in a safe, the robot must switch contact points during manipulation due to the lock state ambiguity, preventing it from determining the sequence of contact points at the outset.

Table 2 compares the richness and diversity of mechanisms between our environment and others. We visualize the adaptive manipulation mechanisms in Figure 2 and Figure 4. More details can be found in Appendix C.2 and D.

## 4 METHOD

As illustrated in Figure 3, we propose a novel framework that learns an adaptive manipulation policy for various mechanisms from collected adaptive demonstrations. To achieve this, we leverage the annotated part poses in our dataset to generate expert manipulation trajectories in the environments, considering invisible internal states to ensure the trajectories are adaptive. Next, to model the expert trajectory distribution with high multi-modality, we employ 3D visual diffusion-based imitation learning (Ze et al., 2024; Chi et al., 2023), which learns the gradient of the action score function to generate actions.

### 4.1 ADAPTIVE DEMONSTRATION COLLECTION

Our goal is to generate adaptive demonstrations that are optimal under partial observation. For instance, when opening a microwave, the expert adaptive policy initially pulls the handle to check the lock state. If the latch is locked, the policy will push the button before opening the door. If the latch is not locked, the policy continues pulling the handle to open the door. In contrast, a static policy

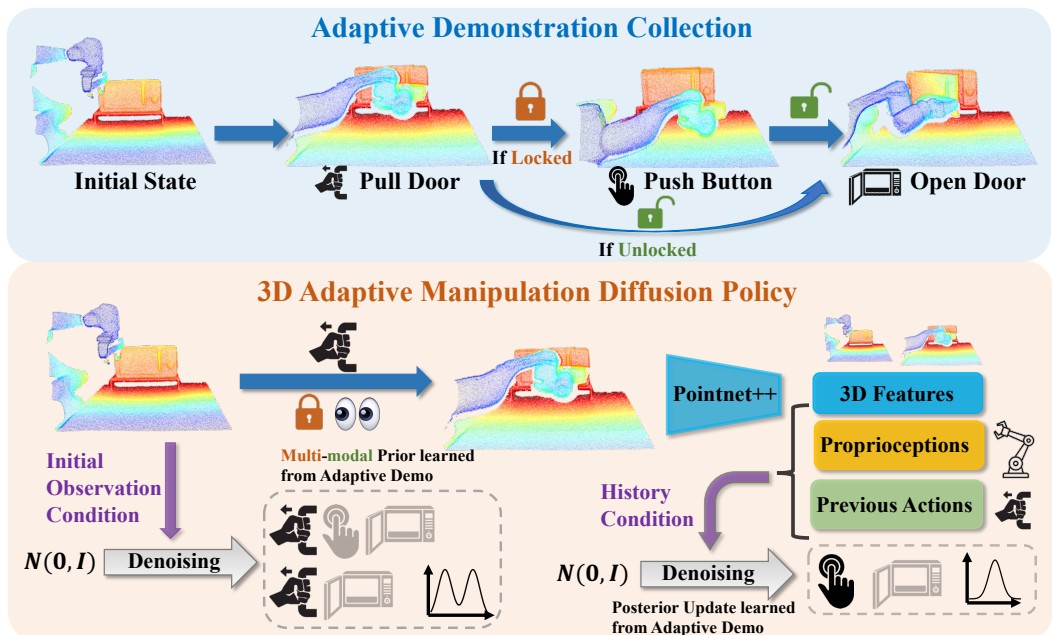

Figure 3: **Adaptive demonstration collection**: Given the uncertain lock state of a microwave, we instruct the robot to first pull the door to check if it is locked, and then follow two different trajectories based on the result. **Diffusion-based 3D adaptive manipulation policy**: Conditioning on the history of 3D visual features, proprioceptions, and actions, the policy denoises Gaussian noise into the trajectory distribution. Initially, the policy captures the bimodal distribution in the demonstration based on the initial observation. As the observed lock state is determined, the policy distribution adaptively shifts to an unimodal distribution.

with full observation would know the lock state in advance and could directly open the door or push the button without first trying to pull the handle.

We design rule-based expert adaptive policies to gather adaptive demonstrations in our simulation environments. We start by computing the bounding box of the part mesh and annotating the part pose through an interactive script. Using these annotations, we label the sequences of parts and manipulation actions for each category, ensuring optimal trajectories under partial observation. For example, the manipulation sequence for a locked safe is grasping the handle, pulling the door, grasping the knob, rotating the knob, grasping the handle, and opening the door. Note that the recorded trajectories are **end effector poses** instead of the high-level action labels. More details are provided in Appendix C. With these policies, we collect adaptive demonstrations of robot motion trajectories for all 9 categories of objects.

## 4.2 3D DIFFUSION-BASED ADAPTIVE POLICY LEARNING

Given the collected demonstration dataset $D = \{(o_t, a_t)\}$, we aim to learn a policy that models the conditional distribution $P(A_t|O_t, \hat{A}_t)$. Here $A_t$ refers to the predicted action sequence $A_t = (a_t, ..., a_{t+T_a})$, where $T_a$ is the action horizon. $O_t$ refers to the observation history, including 3D point clouds and proprioception states, $O_t = (o_{t-T_o}, ..., o_t)$, where $T_o$ is the history horizon. $\hat{A}_t$ refers to action history, $\hat{A}_t = (a_{t-T_o-1}, ..., a_{t-1})$.

However, conducting imitation learning on $D$ is challenging due to its multi-modal nature: The ambiguity of the internal states of articulated objects results in multiple successful manipulation trajectories under the same visual observation. Thanks to recent progress in diffusion-based methods (Chi et al., 2023; Ze et al., 2024; Ke et al., 2024), we can better fit the multi-modal distribution by learning the action score function.

Following the Diffusion Policy (Chi et al., 2023), we utilize DDPM (Ho et al., 2020) to estimate the conditional distribution $P(A_t|O_t, \hat{A}_t)$. The DDPM scheduler performs $K$ iterations of denoising

steps to transform Gaussian noise $A_t^K$ into a noise-free action $A_t^0$. This process adheres to Eq. 1, where $\alpha_k$, $\gamma_k$, and $\sigma_k$ are parameters determined by the scheduler. This reverse process conditions the action prediction on both observations and previous actions, differing from the vanilla implementation of the diffusion policy.

$$A_t^{k-1} = \alpha_k \left( A_t^k - \gamma_k \epsilon_\theta(A_t^k, O_t, \hat{A}_t, k) \right) + \mathcal{N}(0, \sigma_k^2 I) \tag{1}$$

The noise prediction network $\epsilon_\theta$ is trained by minimizing the loss function in Eq. 2, which effectively minimizes the variational lower bound of the KL-divergence between the data distribution and the sample distribution drawn from DDPM:

$$\mathcal{L}(\theta) = \mathbb{E}_{\substack{k \sim \mathcal{U}(0,100), \\ \epsilon^k \sim \mathcal{N}(0, \sigma_k^2 I)}} \left[ \left\| \epsilon^k - \epsilon_\theta(A_t^0 + \epsilon^k, O_t, \hat{A}_t, k) \right\|_2^2 \right] \tag{2}$$

In practice, the observation $o_t$ consists of an observed third-view partial point cloud and the robot's proprioception state (*e.g.*, end-effector pose, robot joint angles, and velocities). The point cloud is first cropped and downsampled by Farthest Point Sampling (FPS) and then encoded by PointNet++ (Qi et al., 2017). The action $a_t$ is the next robot end-effector goal pose. Notably, we find that employing a 6D rotation representation (Ke et al., 2024) for the end-effector pose action stabilizes the training process. During execution, the policy only executes a sub-sequence of the predicted actions, as the adaptive manipulation process requires high-frequency adjustments.

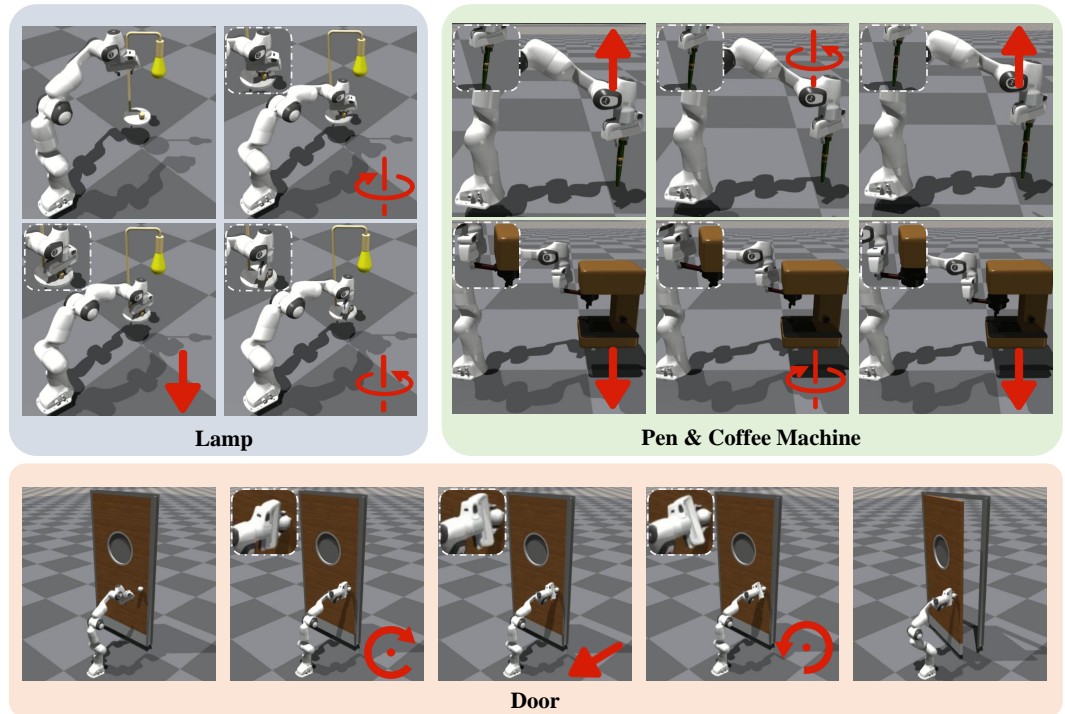

Figure 4: **Adaptive Environments and Qualitative Manipulation Results.** This figure shows the manipulation results of object categories apart from Figure 2.

# 5 EXPERIMENTS

## 5.1 SETTINGS AND METRIC

We conduct experiments in the category level covering all the 9 object categories, and collect 20 adaptive manipulation demonstrations for each object as the training data. For evaluation metric, we use success rate of manipulations. To evaluate what kind of adaptive demonstrations is the most beneficial to adaptive policy learning, we train adaptive policies on adaptive demonstrations with different numbers of adaptive trials.

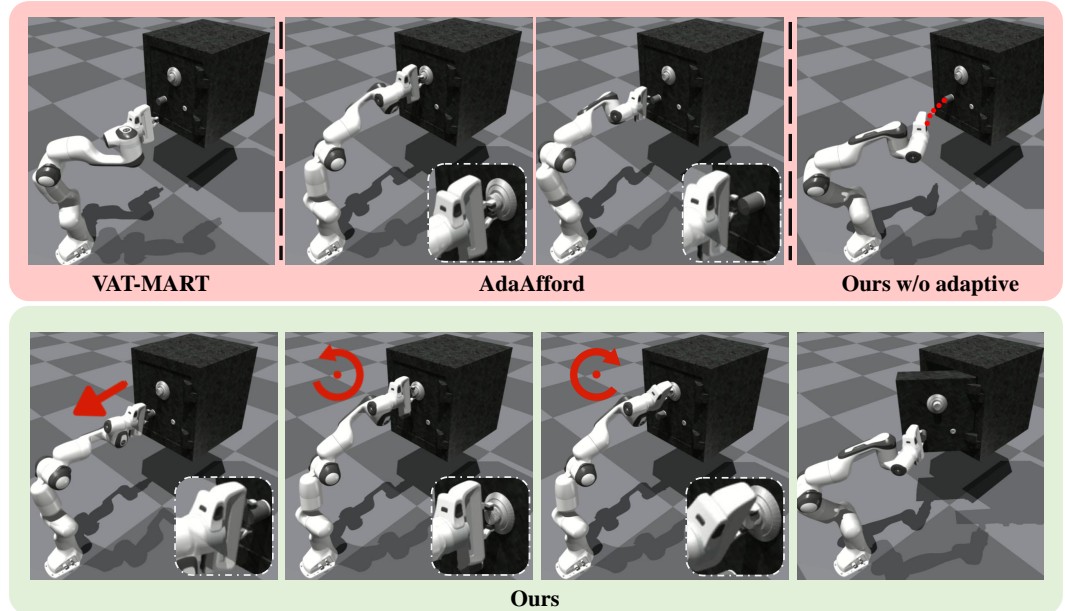

Figure 5: **Manipulation Trajectories Proposed by Our Method and Others.** Our method can sequentially propose stable and accurate adaptive actions, while others have their respective drawbacks.

Table 3: **Success Rates of Different Methods.** Our method outperforms baseline methods and the ablated version in all categories.

| Task | Adaptive Manipulation | | | | | | | | |
|---|---|---|---|---|---|---|---|---|---|
| Method | 🧴 | ✏️ | 🍲 | 🏫 | 📷 | 📠 | ▦ | ▯ | 💡 |
| VAT-MART | 41.43±17.44 | 45.00±12.04 | 38.33±18.33 | 71.43±14.29 | 1.25±0.88 | 6.00±5.10 | 13.75±6.73 | 15.00±12.24 | 34.29±19.38 |
| AdaAfford | 42.86±14.29 | 70.00±11.83 | 55.00±15.00 | 61.43±22.18 | 21.25±16.31 | 77.00±17.35 | 45.00±17.85 | 21.25±11.25 | 52.86±18.13 |
| Sampling | 17.37±9.72 | 24.00±11.14 | 26.67±16.99 | 38.57±16.96 | 15.71±11.87 | 25.00±10.25 | 18.57±12.86 | 11.25±8.75 | 28.75±13.75 |
| ACT | 75.79±8.55 | 74.00±16.85 | 81.11±5.49 | 90.48±6.73 | 28.57±15.43 | 59.00±14.46 | 66.67±5.39 | 52.50±14.58 | 51.19±13.62 |
| DP3 | 83.16±12.19 | 83.00±11.87 | 86.67±4.08 | 85.71±10.10 | 35.71±21.02 | 62.00±13.27 | 70.95±3.37 | 58.75±14.84 | 53.57±13.20 |
| Ours w/o adaptive | 87.14±14.91 | 80.00±10.00 | 85.00±8.90 | 91.42±11.42 | 38.75±20.50 | 58.00±38.16 | 74.77±2.81 | 56.25±13.98 | 54.29±20.00 |
| **Ours** | **95.07±9.70** | **99.05±0.81** | **98.33±4.99** | **97.14±5.71** | **61.25±33.28** | **100.00±0.0** | **94.33±2.38** | **88.75±10.38** | **82.53±14.72** |

## 5.2 BASELINES AND ABLATION

To demonstrate the superiority of our proposed method and its components in manipulating articulated objects with different mechanisms, we compare it with state-of-the-art affordance-based methods, offline imitation learning methods, a sampling-based method, and an ablation of our method.

- **VAT-Mart** (Wu et al., 2022), affordance-based (Zhao et al., 2022; Wu et al., 2023b) method that predicts the open-loop trajectories from the one-frame passive observation.

- **AdaAfford** (Wang et al., 2022), adaptive method that adjusts the manipulation policy based on history interactions using CVAE (Sohn et al., 2015).

- **Sampling**, a planning-based method that samples a macro action from a discrete set at each high-level time step and then plans to the sub-goal pose associated with the selected macro action computed based on the object part pose annotation.

- **ACT** (Zhao et al., 2023), an imitation learning method that uses Action Chunking with Transformers and CVAE.

- **DP3** (Ze et al., 2024), Diffusion Policy extended with 3D visual representations.

- **Ours w/o adaptive**, replace the adaptive demonstration data with static demonstration data that is optimal under full observation, maintaining the diffusion-based imitation learning.

Table 4: **Effects of Repeated Adaptive Trials in Adaptive Demonstration for Bottle.** While adaptive trials are necessary for learning the adaptive policy, repeated adaptive trials make it more difficult to model the adaptive manipulation distribution.

| Trials | 0 | 1(ours) | 2 | 3 | 4 | 5 | 6 |
|---|---|---|---|---|---|---|---|
| **Success Rate** | 0.8714 | **0.9364** | 0.7571 | 0.6571 | 0.5714 | 0.4143 | 0.4428 |

## 5.3 SIMULATION RESULTS

Table 3 and Figure 5 respectively show the success rates and manipulation trajectories of different methods. Figure 4 further shows more manipulation trajectories proposed by our method. Our method outperforms all other methods and demonstrates stable and accurate action trajectories for adaptive manipulation. From the visualizations, we can observe that, for **VAT-Mart**, as an open-loop method, it is difficult to fit the whole trajectory space and directly predict a manipulation trajectory at a time, and the open-loop method does not support adapting the policy from previously executed actions. Using the diffusion-based imitation method, our method can more accurately model the poses of actions using the limited number of data (such advantage is also demonstrated in other diffusion-based imitation studies (Chi et al., 2023; Ze et al., 2024)). In contrast, **AdaAfford** suffers from the inferior multi-modal distribution modeling capability of CVAE and the increased training data requirements associated with point-level affordance learning. The **Sampling**-based method performs worse than AdaManip and other baselines because its sampling process because its sampling process cannot efficiently leverage priors or posteriors learned from demonstrations. While **ACT** and **DP3** outperform affordance-based methods, they fall short of AdaManip due to the absence of an adaptive demonstration collection pipeline. Additionally, **ACT** is further limited by its lack of the robust multi-modality modeling capability provided by diffusion models.

**Ours w/o adaptive** is not trained from demonstrations with recovery actions from failure trials, so the learned policy could not adapt from failure actions to finally achieve the goal. As illustrated in Figure 5, when the safe is locked and **Ours w/o adaptive** attempts to open the door, it fails to switch to rotating the knob and instead continues on the opening trajectory.

Table 4 shows the effects of repeated adaptive trials in adaptive demonstrations for training. When only using optimal successful actions (*i.e.*, 0 adaptive trials) under full observation, the model is the same with **Ours w/o adaptive** and could not have the adaptation capability. When using more than one adaptive trial at the same object state, these adaptive trials are redundant and increase the complexity of distributions to model, while not increasing the scenarios the policy can handle. Therefore, the performance decreases when the number of repeated adaptive trials increases from 1. These results validate our demonstration collection design, which limits the manipulation sequence to only one adaptive trial.

Table 5: Real-world Evaluation Results.

| Tasks | Open Bottle | Open Microwave | Open Safe | Open Pressure Cooker |
|---|---|---|---|---|
| **Success** | 8/10 | 7/10 | 5/10 | 5/10 |

## 5.4 REAL-WORLD EXPERIMENTS

To validate the generalization of our adaptive diffusion policy to real-world scenarios, we conduct experiments on various real-world objects like Pressure Cooker, Microwave, Bottle, and Safe. For the real-world settings, we employ a Franka Emika Panda Robot Arm as our agent. To capture 3d visual observation, we position an Azure Kinect DK camera adjacent to the robot arm. Our policy takes the real-time point clouds from the depth camera, the robot state from the robot arm, and previous actions as the input, and generates the corresponding end-effector action in a close-loop fashion. We collected 35 adaptive expert demonstrations for each object by human teleoperation to train the policies, and conducted 10 evaluation trials per object.

Table 5 presents the number of successful executions across four real-world tasks. The results in Figure 6 demonstrate that our adaptive policy can be effectively applied to real-world scenarios. To

better illustrate this adaptive behavior, we visualize different trajectories under different object states in Figure 7. The red trajectory shows that the robot finds out that the microwave is locked after it failed to pull the door and turned to push the button to unlock the door. Conversely, if the microwave is initially unlocked, the robot will continue to open the door after it grasps the handle, as depicted by the blue trajectory in sub-figure 2 of Figure 7. For more visualizations, please see Appendix D.

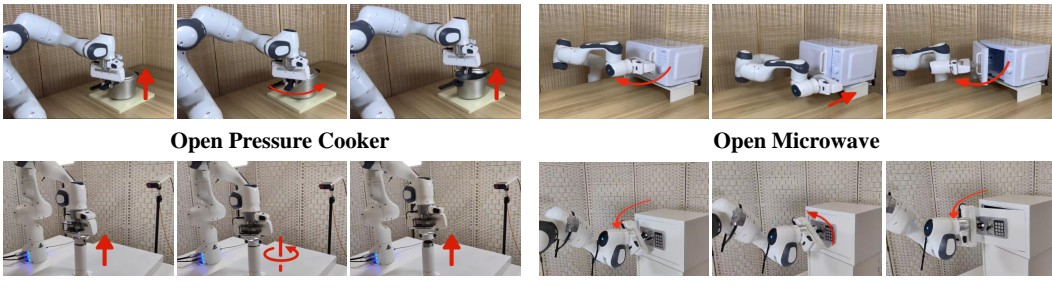

Figure 6: Manipulation Trajectories of Real-World Scenarios.

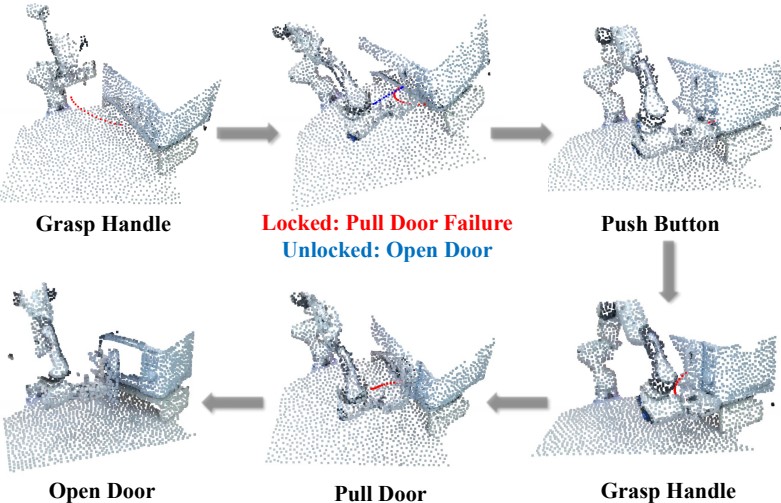

Figure 7: Visualization of adaptive policy of Open Microwave in the real world. The trajectories vary depending on the state of the microwave. The red trajectory represents the executed trajectory when the microwave is locked. In subfigure 2, the blue trajectory illustrates the robot's action when the microwave is unlocked, prompting it to continue opening the door. The red trajectory shows that the robot failed to pull the locked door and turned to push the button.

## 6 CONCLUSION

We study the problem of adaptive manipulation policy for manipulating articulated objects with diverse and complex mechanisms, build environments with different categories of such objects that support the various manipulation mechanisms, and propose a novel framework that learns the adaptive manipulation policy for various mechanisms from diverse adaptive demonstrations based on diffusion policy. The significance of our proposed environment and the effectiveness of the proposed adaptive policy learning framework have been demonstrated by our experiments.

Our paper represents the initial study into the environment suitable for adaptive manipulation policy learning. Currently, our dataset encompasses 9 categories with a total of 277 objects, which we plan to expand by introducing more categories and instances to cover increasingly complex and realistic mechanisms. Additionally, incorporating deformable object manipulation into adaptive tasks represents a significant direction for future research. Moving forward, the AdaManip environment will progressively include a broader spectrum of real adaptive manipulation tasks, making it a comprehensive platform for both training and testing adaptive manipulation policies.

## ACKNOWLEDGEMENTS

This project is supported by the National Science and Technology Major Project (2022ZD0114904), the National Natural Science Foundation of China (No. 62376006), the National Youth Talent Support Program (8200800081), the National Natural Science Foundation of China (No. 62136001) and NSFC-6247070125.

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

Table 6: **Parameters for training and diffusion model**

| Training | Values |
|---|---|
| Hardware Configuration | NVIDIA GeForce GTX 4090 |
| Weight Decay | 1e-6 |
| Batch Size | 64 |
| Optimizer | Adam |
| Learning Rate | 1e-4 |
| Epochs | 500 |
| Time Expense | 3h |
| **Model** | **Values** |
| Pointcloud Size | 4096 |
| Backbone | Unet |
| Observation History Horizon | 4 |
| Prediction Horizon | 4 |
| Action Horizon | 2 |
| EMAModel | True |
| Diffusion Timestep | 100 |
| Noise Scheduler | squaredcos |
| Action Space | absolute end-effector pose |

## A  HYPER-PARAMETERS FOR EXPERIMENTS

Table 6 summarizes the hyperparameters for our diffusion policy model and training details.

## B  DATASET CONSTRUCTION

Apart from object assets collected from existing datasets (Xiang et al., 2020; Li et al., 2024b), most of the object assets in our dataset are obtained from 3Dwarehouse (Trimble). We dedicate significant time and effort to carefully selecting the available object meshes, segmenting them into distinct parts, re-aligning the object mesh coordinate systems, and subsequently developing Python scripts to facilitate the efficient synthesis of operational dataset instances. Similar to GAPartNet (Geng et al., 2023b), we annotate the object assets with rich and comprehensive labels.

## C  DETAILED EXPERT POLICY DESIGN FOR ADAPTIVE DEMONSTRATION COLLECTION

### C.1  POSE ANNOTATION

To achieve precise part operation, we employ a pose labeling approach for the parts. Initially, we roughly estimate the part pose through the bounding box of the part mesh calculated using the Python library trimesh. Subsequently, we observe the pose annotation results in real-time through an online interactive script to complete the precise annotation of the pose.

### C.2  ADAPTIVE MANIPULATION SEQUENCE

**Bottle:** Grasp the Cap. Rotate Cap after a failed Lift Up. Randomly sample Rotate/Lift if the previous action is Rotate.

**Pen:** Same as Bottle.

**Pressure Cooker:** Grasp the Handle. Rotate Handle after a failed Lift Up. Randomly sample Rotate/Lift if the previous action is Rotate.

**Coffee Maker:** Grasp Portafilter. Rotate Portafilter after a failed Pull Down. Randomly sample Rotate/Pull if the previous action is Rotate.

**Window:** Grasp the Handle. Randomly choose a direction to Rotate Handle. If failed, choose the other direction. Rotate Handle after a failed open trial. Randomly sample Rotate/Open if the previous action is Rotate.

**Door:** Same as Window.

**Lamp:** Randomly choose to Push/Clockwise Rotate/Counter Clockwise Rotate. Never choose a failed action.

**Safe:** Pull Door. If succeed, then continue opening the door. If failed, Rotate Knob. Randomly choose a direction to Rotate and choose the other one if failed. Then Pull Door again to open it.

**Microwave:** Pull Door. If succeed, then continue opening the door. If failed, Push Button. Then Pull Door again to open it.

### C.3    TRAJECTORY SPARSIFICATION

If the history consists of a dense trajectory of the robot's end effector poses, the policy would require a long history context length to capture previous failures. However, training a policy with a long history context is challenging, as it requires more computing resources and is less robust. To mitigate this issue, we chose to sparsify the trajectory. Only key frames of the demonstration trajectories are saved for imitation learning, but the recorded actions are still 6D end effector poses instead of high-level macro actions.

For example, in the open-safe task, the recorded history includes grasping poses and several manipulation poses while omitting most intermediate steps. The history condition for the policy is as follows: [grasp the handle, pull the door and fail to open]. Other intermediate poses during execution are excluded from the history. Based on this context, the robot predicts the next goal pose [unlock the key]. Once the goal pose is predicted, we apply inverse kinematics (IK) to plan the path for execution.

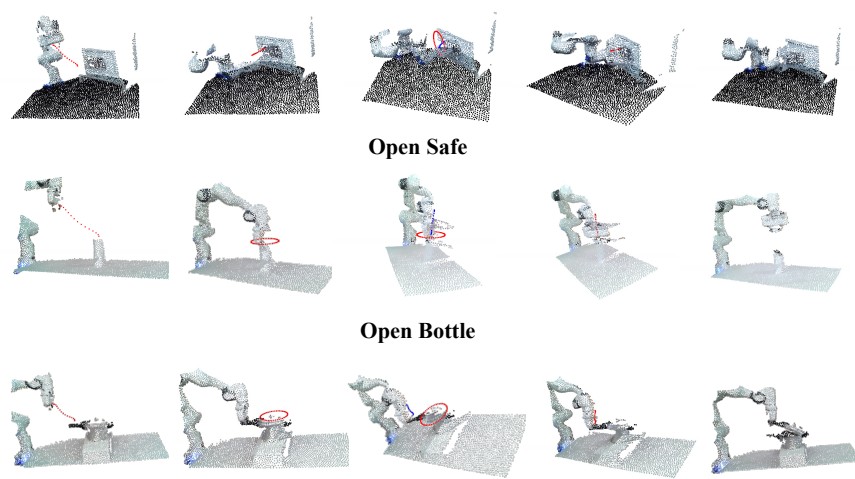

Figure 8: Visualization of Open Safe/Bottle/Pressure Cooker Experiment in the real world. The red trajectory represents the executed trajectory. The blue trajectory indicates the actions under other object states.

## D    MECHANISM VISUALIZATION AND EXPERIMENTAL RESULTS

Figure 4 illustrates the operating mechanism of the remaining categories and the experimental results. The mechanism of the lamp in the upper left corner is **"Push/Rotate"**. which can be operated by pressing/rotating the button/knob, but only one way is correct. The pen and coffee machine in the

upper right corner fall under the **"Rotate/Slide"**. To operate them, you need to rotate a specific part to a certain angle to open the pen cap or the coffee machine handle. The door at the bottom belongs to the **"Random Rotation Direction"** and **"Lock"**, where the handle must be rotated either clockwise or counterclockwise to open the door.

Figure 8 shows the visualizations of the other 3 experiments in real world apart from Open Microwave shown in Figure 7. In these visualizations, the red trajectories represent the paths actually executed by the robot, while the blue trajectories indicate potential paths under different object states. For instance, in Sub-figure 3 of the Open Bottle experiment, the blue trajectory shows that if the cap is rotated sufficiently, the robot can then lift the cap. This visualization helps illustrate the adaptability of the robot's actions based on the observed state of the objects involved.

