# OpenReview forum: "AdaManip: Adaptive Articulated Object Manipulation Environments and Policy Learning"
_ICLR.cc/2025/Conference — ICLR 2025 Poster_

### Official Review · Reviewer_RR8i · 2024-11-02

**Soundness:** 3
**Presentation:** 3
**Contribution:** 2
**Rating:** 5
**Confidence:** 5

**Summary:**

This paper presents a novel robotic simulation environment and a dataset which targets solving articulated object manipulation tasks with some adaptive mechanisms, adjusting the feedback from failed or successful past actions. The approach uses a 3D diffusion-based behaviour cloning method to model multi-modal action distribution. The experiments show that this method outperforms existing models in both simulated and real-world scenarios, illustrating its potential for real-world applications.

**Strengths:**

1. The paper introduces a new simulation setup with more varied and complex object interactions, which improves over existing datasets. This new setup helps robots learn and practice in a way that better matches real-world conditions, making their training and testing more realistic.
2. The presented experimental results demonstrated the proposed policy outperformed the baseline on a significant level, proving its effectiveness and applicability.
3. Real-world experiments are promising and impressive.

**Weaknesses:**

1. The paper's main contributions focus on introducing a new manipulation environment and demonstrating real robot experiments, but it lacks significant innovation in the underlying methodology, raising questions about its fit for this particular conference.
2. The paper lacks comprehensive ablation experiments to justify design choices. For example, it does not explain why action history was chosen over observation history, which is more common in reinforcement learning research. Additionally, there is no comparison with sampling-based methods that first identify plausible actions and then plan over them.
3. The paper does not discuss the scalability of the approach or its ability to generalize across different objects in a zero-shot setting. It also fails to explore how the method would handle combinations of the five proposed manipulation mechanisms.
4. The method relies heavily on the quality of the collected data, and the diffusion model learns everything simultaneously, which could lead to redundancy and potential overfitting to the training data.

**Questions:**

1. How does your method provide significant methodological innovation beyond creating a novel environment and conducting real robot experiments?
2. Why did you choose to use action history instead of observation history, which is more commonly used in reinforcement learning? Have you conducted experiments to compare the effectiveness of these approaches?
3. How does your method perform compared to sampling-based methods that propose plausible actions and plan over them? Have you evaluated this as a baseline?
4. Can you elaborate on how your approach scales to new, unseen objects in a zero-shot setting? What is the expected performance when combining multiple proposed mechanisms?
5. How do you ensure that your method generalizes effectively across different categories of articulated objects?
6. Could you provide more details on how your method handles potential overfitting, given its reliance on the quality of collected data and the "learn everything at once" nature of the diffusion model?
7. What are the limitations of your adaptive policy learning approach when applied to more complex or real-world scenarios with more diverse mechanisms?

---

> ### Author Response · Authors · 2024-11-22
> **Response to Reviewer RR8i [Part 1/3]**
>
> Thank you for your valuable review! We would like to address your concerns in the following response.
>
> **Q1: The paper's main contributions focus on introducing a new manipulation environment and demonstrating real robot experiments, but it lacks significant innovation in the underlying methodology, raising questions about its fit for this particular conference.**
>
> We would like to emphasize that the contribution of our paper includes both the introduction of the articulated object manipulation simulation and dataset, and the proposed whole learning framework for adaptive policies based on the simulation. Adaptive articulated object manipulation represents a set of realistic and frequently encountered tasks for robots, while relatively underexplored due to the few attempts to build the simulation environment and dataset supporting the adaptive policy training. As recognized by Reviewer XbhV and eEiz, our work introduces **the adaptive articulated object manipulation task** along with the corresponding **AdaManip environment and dataset**. Moreover, we design an **adaptive demonstration collection and policy learning pipeline**, enabling the policy to model multi-modal trajectory distributions and update its posterior based on environmental feedback.
>
> Through this work, we aim to pave the way for further research on real-world manipulation challenges, and highlight the importance of learning the efficient adaptive manipulation policy by collecting adaptive demonstrations over optimal demonstrations.
>
> Regarding the fit for this conference, we believe our work strongly aligns with the accepted topic of "applications to robotics, autonomy, planning" at ICLR. Furthermore, recent ICLR papers, including ManiSkill2 [1] and DaxBench [2] (notable-top-5% at ICLR 2023), have focused on proposing realistic robot simulation environments without necessarily introducing significant innovations in learning algorithms. The contributions of our work are of a similar scope, providing a valuable resource for advancing robotics research, with a further designed learning framework in this novel environment.
>
>
> **Q2: Why action history was chosen over observation history, which is more common in reinforcement learning research.**
>
> As we mentioned in Sec 4.2, AdaManip policy "models the conditional distribution $P(A_t|O_t,\hat{A}_t)$. Here $A_t$ refers to the predicted action sequence, $O_t$ refers to the observation history, including 3D point clouds and proprioception states, and $\hat{A}_t$ refers to action history." Therefore, the policy is conditioned on **both observation history and action history**.
>
> **Q3: The generalization ability of this method.**
>
> The testing is conducted on the training shapes at the category level, with variations in the object positions, poses (orientations), and joint configurations in both simulation and real-world experiments. Such setting aligns with current imitation learning studies [3, 4] using few-shot demonstrations (in our work, we use only 20 demos per shape for training).
>
> As for the generalization towards novel shapes, another branch of current works is trying to use large-scale data [5] or foundation models [6] to achieve such generalization capability, which is orthogonal to the contribution to our work: empowering imitation learning with the adaptive manipulation capability on diverse object mechanisms with our proposed environment, mechanisms and assets.
>
> It is a promising direction for the following work to further study how to leverage the generalizable knowledge inherited from large models to enhance the cross-shape and cross-category generalization capability of adaptive manipulation on diverse object mechanisms.
>
> [1] Gu et al, Maniskill2: A unified benchmark for generalizable manipulation skills, ICLR 2023
>
> [2] Chen et al, Daxbench: Benchmarking deformable object manipulation with differentiable physics, ICLR 2023
>
> [3] Chi et al, Diffusion Policy: Visuomotor Policy Learning via Action Diffusion, RSS 2023
>
> [4] Zhao et al, Learning Fine-Grained Bimanual Manipulation with Low-Cost Hardware, RSS 2023
>
> [5] Yu et al, RT-1: Robotics Transformer for real-world control at scale, ArXiv 2022
>
> [6] Yan et al, DNAct: Diffusion Guided Multi-Task 3D Policy Learning, ArXiv, 2024

---

> > ### Comment · Reviewer_RR8i · 2024-11-26
> > **Rebuttal Response**
> >
> > Thank you for the clarification. While I see the value in highlighting differences, I feel it may not be entirely appropriate to compare this work directly with ManiSkill2 and DaxBench. Those works focus on developing new simulators or physics engines, such as GPU parallelization in ManiSkill2 and simulator differentiability in DaxBench. In contrast, this work creates a benchmark by composing articulated objects in IsaacGym using PartNet assets. Additionally, the paper does not explicitly position itself as a benchmark paper, which sets it apart from these prior works.

---

> > > ### Author Response · Authors · 2024-11-27
> > > **Second Round Response to Reviewer RR8i [Part 1/2]**
> > >
> > > Thank you for your thoughtful feedback. As you noted, our work is not positioned as a benchmark paper. This is because our contributions extend beyond the development of new object assets and environments—we also propose a complete pipeline for collecting adaptive demonstrations and training manipulation policies. In contrast, ManiSkill2 and DaxBench primarily focus on advancing simulators or physics engines. We referred to these papers in our previous response to emphasize that introducing a new dataset and environment represents a significant contribution recognized by the robotics community at ICLR. This was intended to address concerns about the fit of our work for this conference.
> > >
> > > Additionally, we would like to clarify that the object assets in our AdaManip environment are **NOT** directly borrowed from the PartNet dataset. In fact, most of the asset materials are obtained from 3D Warehouse. We dedicated significant time and effort to carefully selecting the available object meshes, segmenting them into distinct parts, re-aligning the object mesh coordinate systems, and subsequently developing Python scripts to facilitate the efficient synthesis of operational dataset instances. Thus, both the new assets and the adaptive mechanism simulation environment are integral contributions of the AdaManip dataset.

---

> ### Author Response · Authors · 2024-11-22
> **Response to Reviewer RR8i [Part 2/3]**
>
> **Q4: How would the method handle combinations of the five proposed manipulation mechanisms?**
>
> To demonstrate the capability in handling mechanism combinations, we show how our method tackles the open-safe task, the most complicated task in the current version of the AdaManip environment, which incorporates **the combinations of three of the five** mechanisms: the Lock Mechanism, Random Rotation Direction, and Switch Contact Mechanism. In this task, the robot needs to determine whether the door is locked, the rotation direction of the knob lock, and the correct contact point to reach.
>
> To handle this task, we designed a rule-based adaptive demonstration collection policy to cover all possible failure and success states, enabling the policy to adjust its actions based on feedback. Specifically, the demonstration policy follows these steps: 1. The robot first pulls the door handle. If successful, it proceeds to open the door. 2. If pulling the handle fails, the robot switches to rotating the knob. 3. It randomly chooses a rotation direction and switches to the other direction if the first attempt fails. 4. Once the knob is successfully rotated to unlock the door, the robot pulls the handle again to complete the task.
>
> In the future, we plan to continually introduce more categories and mechanisms into the AdaManip environment. This will include more challenging tasks that combine multiple mechanisms, increasing complexity and ambiguity. Similarly, for those tasks, we can design above-described demo collection policy accordingly or collect adaptive demo by human teleoperation, and then learn policies covering mechanism combinations using the collected data.
>
>
> **Q5: The method relies heavily on the quality of the collected data, and the diffusion model learns everything simultaneously, which could lead to redundancy and potential overfitting to the training data.**
>
> The training data encompasses both failure and success states encountered during object manipulation, which prevents overfitting to specific successful trajectories. Additionally, it incorporates **optimal recovery** actions conditioned on failures, ensuring redundancy is avoided. The results in Table 4 further validate the efficiency of our scripted policy in minimizing redundant adaptive trials.
>
> The diffusion model, trained on the adaptive and efficient demonstrations, captures the multi-modal prior distribution and develops the ability to adjust its posterior based on feedback. This training approach prevents overfitting to static trajectories and enables the model to dynamically adapt its behavior based on historical context.
>
> For example, in the open-safe task, the policy initially predicts multiple possibilities, such as directly opening the door or rotating the knob to unlock it. If the robot attempts to pull the handle but fails to open the door, the policy updates its posterior based on this feedback, collapsing the multi-modal prediction into a single mode—rotating the knob. By learning from adaptive demonstrations and utilizing failure feedback, the AdaManip model effectively avoids overfitting to static trajectories and redundant trials.

---

> > ### Comment · Reviewer_RR8i · 2024-11-26
> > **Rebuttal Response**
> >
> > Thank you for the detailed response. I appreciate the explanation of the rule-based demonstration policy and how it enables adaptability. However, I am curious about how the robot identifies the affordance of a knob to rotate, determines the correct axis of rotation, and selects the direction. Are these affordances explicitly encoded in the expert training data, or does the model autonomously infer them during training? If they depend heavily on the quality of expert demonstrations, it would be helpful to understand how these demonstrations are structured to ensure the robot learns such affordances effectively across diverse tasks and objects.

---

> > > ### Author Response · Authors · 2024-11-27
> > > **Second Round Response to Reviewer RR8i [Part 2/2]**
> > >
> > > Thank you for your insightful question. We appreciate the opportunity to provide further clarification regarding the policy's output space and training/inference process.
> > >
> > > The AdaManip policy predicts 6D pose action sequences (position and orientation) for execution. To train this policy, we leverage the rule-based demonstration policy, which generates 6D pose action sequences using object part pose annotations (e.g., the pose of a bottle cap). These annotations guide the demonstration policy to provide high-quality adaptive demonstrations tailored to the task requirements.
> > >
> > > The policy is trained end-to-end using supervised learning, directly guided by the 6D pose action sequences from the rule-based demonstrations. During inference, the trained policy outputs the 6D pose actions for execution. Notably, this approach eliminates the need for explicit affordance inference at runtime, as the training process ensures that the policy internalizes the required actions and adaptations based on the demonstration data.

---

> ### Author Response · Authors · 2024-11-22
> **Response to Reviewer RR8i [Part 3/3]**
>
> **Q6: How does your method perform compared to sampling-based methods that propose plausible actions and plan over them? Have you evaluated this as a baseline?**
>
> Thank you for your suggestion! We implemented a sampling-based method as an additional baseline. Specifically, the policy samples a macro action from a discrete set at each high-level time step and then plans to the sub-goal pose associated with the selected macro action.
>
> For example, in the open-safe task, the macro action set is {grasp the door handle, grasp the knob, rotate the knob clockwise, rotate the knob counterclockwise, pull the door}. The sub-goal pose for each macro action is computed based on the object part pose annotation.
>
> The quantitative results for this sampling-based method are shown in the following table. This method performs worse than AdaManip and other baselines. The main reason is that the sampling process lacks efficient way in leveraging prior or posterior learned from demonstrations.
>
> | **Method**    | **Bottle**         | **Pen**             | **PressureCooker**  | **CoffeeMachine**  | **Safe**            | **Microwave**       | **Window**          | **Door**           | **Lamp**            |
> | --- | -------------- | --------------- | --------------- | --------------- | --------------- | --------------- | --------------- | -------------- | --------------- |
> | Sampling   | 17.37$\pm$9.72 | 24.00$\pm$11.14 | 26.67$\pm$16.99 | 38.57$\pm$16.96 | 15.71$\pm$11.87 | 25.00$\pm$10.25 | 18.57$\pm$12.86 | 11.25$\pm$8.75 | 28.75$\pm$13.75 |
>
>
> **Q7: What are the limitations of your adaptive policy learning approach when applied to more complex or real-world scenarios with more diverse mechanisms?**
>
> Currently, adaptive demonstrations are collected using a human-designed rule-based policy (in simulation) or through human teleoperation (in real-world experiments). While this approach is commonly leveraged in other works (e.g., [7], which also use rule-based policies for simulation data collection and human teleoperation for real-world data collection) and has successfully facilitated adaptive policy learning, these data collection pipelines require significant human effort for each task. As the complexity and diversity of mechanisms increase, designing such pipelines for each category may become impractical. A promising direction for future work would be to explore automated or streamlined methods for data collection.
>
> Additionally, as the task complexity increases, the policy will need to retain a longer history context to accurately identify and recover from failures. This increases both the computational cost for training and inference and may also introduce instability during training due to the challenges of long-context modeling.
>
> [7] Wang et al, GenDP: 3D Semantic Fields for Category-Level Generalizable Diffusion Policy, CoRL 2024

---

> ### Author Response · Authors · 2024-11-29
>
> We sincerely thank you for your thoughtful review. As the discussion period draws to a close, we would appreciate your feedback to confirm whether our replies have addressed your concerns. If you have any remaining questions, we are happy to provide further clarification. If our responses have resolved your concerns, we would be deeply grateful if you could consider raising the score. Thank you again for your time and effort during the review process.

---

> > ### Comment · Reviewer_RR8i · 2024-12-02
> > **Second Round Response to Authors**
> >
> > Thank you for your thoughtful response. Regarding the novelty of this paper, I still find that combining sequential affordances, without proposing a robust or innovative algorithm to target and solve the problem effectively, offers limited contributions. The origins of the assets (whether from PartNet or 3D Warehouse) are not the core issue. Instead, the fundamental novelty of the dataset should lie in meaningful advancements, such as improvements to the physics engine, rigid body collision handling, or ray tracing capabilities. Without such breakthroughs, I believe the dataset's contribution does not sufficiently meet the standards of this conference.
> >
> > Additionally, the action decoder described in the paper appears to heavily rely on supervised data, specifically the rule-based demonstrations you referenced. This dependence results in a relatively standard Behavior Cloning approach, without incorporating specific techniques to address sequential affordance reasoning. Concepts such as causality or affordance discovery, which have broader utility and deeper implications, could have been more thoroughly explored to enhance the methodology's impact.
> >
> > While I acknowledge the authors’ effort and the potential of this research direction to make a meaningful contribution to the community, this particular paper still falls short of achieving the level of novelty and innovation I would expect for acceptance. Thus, I remain inclined to recommend a weak rejection for this submission.

---

> > > ### Author Response · Authors · 2024-12-03
> > >
> > > Thank you for your detailed reply and constructive suggestions! We would like to make a few clarifications in this response.
> > >
> > > First, we would like to clarify that the AdaManip policy is not trained to predict point-level affordances for guiding manipulation. Instead, the policy is directly supervised by the 6D pose trajectories and outputs a sequence of pose actions, rather than sequential affordances.
> > >
> > > Secondly, we believe that the adaptive manipulation problem is both important and under-explored. The core contribution of our work lies in the proposal of this problem and the implementation of the mechanisms to simulate it, rather than in the introduction of a new physics engine or ray tracing, which is not inherently necessary for studying the adaptive mechanisms. The reason we chose not to simulate complex lock collision shapes is that such simulations can introduce uncontrollable errors, such as interpenetration (as discussed in [1], Appendix C), leading to instability. Therefore, we believe the current dataset and environment is an appropriate and valuable resource for studying adaptive manipulation.
> > >
> > > Thirdly, we appreciate your suggestion to explore techniques for sequential affordance reasoning. While our work does not focus on affordance learning, we agree that it would be valuable to investigate how to better capture the causal link between history and adaptation. Currently, AdaManip learns to update the posterior distribution based on interaction history implicitly through end-to-end imitation learning on adaptive demonstrations, which we think can be viewed as a form of "affordance discovery." Since the adaptive manipulation problem is relatively unexplored, we chose the standard imitation learning setting, which is widely accepted in other tasks such as garment manipulation, and we present this results as an initial attempt to solve the adaptive manipulation task. We do believe that exploring causal discovery and online reasoning could further enhance the policy, and we plan to elaborate on these aspects in future work, as they are outside the current scope due to time constraints.
> > >
> > > Thank you once again for your detailed feedback and valuable suggestions!
> > >
> > > [1] Tang et al., Automate: Specialist and generalist assembly policies over diverse geometries, RSS 2024.

---

### Official Review · Reviewer_XbhV · 2024-11-02

**Soundness:** 4
**Presentation:** 4
**Contribution:** 3
**Rating:** 8
**Confidence:** 4

**Summary:**

This work presents a new simulation environment and framework for adaptive manipulation of articulated objects with diverse mechanisms. The AdaManip environment includes nine object categories and five types of *adaptive* mechanisms. Using a novel adaptive collection method, the authors generate a dataset to support training for complex, multi-step manipulation tasks. The paper further introduces a 3D diffusion-based imitation learning framework to model multi-modal action distributions, enabling policies that adjust based on previous actions. Well-designed experiments demonstrate results in simulation and real-world scenarios indicating that AdaManip achieves high success rates and outperforms baseline methods in handling complex articulated objects.

**Strengths:**

- The AdaManip environment addresses a gap in existing robotic manipulation environments by focusing on objects with realistic, multi-step (adaptive) mechanisms, supporting a range of complex manipulation scenarios. This is valuable for advancing robot manipulation learning on real-world objects and increasing robots' learned policies' generalizability.

- Adaptive Demonstration Collection: The method for collecting adaptive demonstration trajectories in a rule-based fashion is well-motivated.

- The use of diffusion models for action distribution in policy learning is innovative and effectively models the multi-modal distribution necessary for tasks where visual cues alone are insufficient.

- Extensive Evaluation: The paper provides detailed simulation and real-world experiments. Performance metrics such as success rates across nine object categories and comparisons with baseline methods, including ablation studies, illustrate the framework's effectiveness.

**Weaknesses:**

- Although the framework outperforms baselines like VAT-MART and AdaAfford, it would benefit from additional baseline comparisons, especially with RL methods (or more specifically, with dense reward annotation) or DAgger, that may be adaptable to adaptive manipulation tasks, or recent work, FlowBotHD, that incorporates history into flow in order to learn to adapt to past errors using a flow-based policy (which is your sequential rule-based policy but with flow representations).

- More emphasis on how AdaManip differs from environments like PartNet-Mobility in complexity or adaptability would better explain the novelty of AdaManip. For example, you could talk about whether there are particular types of mechanisms or object categories that AdaManip includes that PartNet-Mobility does not.

**Questions:**

- Since the policy is a rule-based sequence, how does the proposed method compare with traditional planning methods? Specifically, please provide a brief comparison or discussion of how your method relates to or differs from traditional planning approaches for articulated object manipulation.

- Is diffusion policy the only viable solution? Specifically,  maybe discuss potential alternative approaches and explain why you chose diffusion policy over other methods.

---

> ### Author Response · Authors · 2024-11-22
> **Response to Reviewer XbhV**
>
> Thank you for your insightful feedback and for highlighting the importance of our work. We hope to address your concerns in the response below.
>
> **Q1: More comparison with additional baselines, including RL and traditional planning methods.**
>
> Thank you for your suggestion! In our imitation learning setting, the adaptive policy is trained using a limited number of demonstration trajectories, which aligns with the practical setting adopted by previous methods, such as Diffusion Policy and ALOHA. Therefore, it would not be a fair comparison to evaluate against RL methods that rely on online data collection beyond offline demonstrations.
>
> We implemented a sampling-based method as an additional planning-based baseline. Specifically, the policy samples a macro action from a discrete set at each high-level time step and then plans to the sub-goal pose associated with the selected macro action.
>
> For example, in the open-safe task, the macro action set is {grasp the door handle, grasp the knob, rotate the knob clockwise, rotate the knob counterclockwise, pull the door}. The sub-goal pose for each macro action is computed based on the object part pose annotation.
>
> The quantitative results for this sampling-based method are shown in the following table. This method performs worse than AdaManip and other baselines. The main reason is that the sampling process lacks efficient way in leveraging prior or posterior learned from demonstrations.
>
> | **Method**    | **Bottle**         | **Pen**             | **PressureCooker**  | **CoffeeMachine**  | **Safe**            | **Microwave**       | **Window**          | **Door**           | **Lamp**            |
> | --- | -------------- | --------------- | --------------- | --------------- | --------------- | --------------- | --------------- | -------------- | --------------- |
> | Sampling   | 17.37$\pm$9.72 | 24.00$\pm$11.14 | 26.67$\pm$16.99 | 38.57$\pm$16.96 | 15.71$\pm$11.87 | 25.00$\pm$10.25 | 18.57$\pm$12.86 | 11.25$\pm$8.75 | 28.75$\pm$13.75 |
>
>
> **Q2: More emphasis on how AdaManip differs from environments like PartNet-Mobility in complexity or adaptability would better explain the novelty of AdaManip. For example, you could talk about whether there are particular types of mechanisms or object categories that AdaManip includes that PartNet-Mobility does not.**
>
> Previous articulated object environments do not simulate the complex mechanisms implemented in AdaManip. For example, in PartNet-Mobility, opening a microwave door involves directly pulling it open. In contrast, our environment requires the robot to first push a button to unlock the door; otherwise, the door remains locked and cannot be opened. Similarly, in PartNet-Mobility, a bottle cap can be directly lifted off. In our environment, however, the robot must first rotate the cap to a specific degree before it can be lifted.
>
> All nine categories in our dataset, along with their corresponding mechanisms, are not covered in previous datasets such as PartNet-Mobility. This highlights the complexity and adaptability offered by AdaManip, setting it apart from existing environments.
>
>
> **Q3: Is diffusion policy the only viable solution? Specifically, maybe discuss potential alternative approaches and explain why you chose diffusion policy over other methods.**
>
> **In our imitation learning setting**, the adaptive policy is trained using a limited number of demonstration trajectories, which aligns with the practical setting adopted by previous methods. Adaptive manipulation tasks inherently require the policy to address the ambiguity of manipulation trajectories under pure visual observation. Therefore, the model must excel at modeling multi-modal distributions, and diffusion models are well-known for their strong capability in this regard. The results from the additional ACT experiment further confirm that diffusion-based approaches outperform others in this context.
>
> **Exploring alternative settings**, such as online learning methods (e.g., RL), foundation model based manipulation, or large pretrained visual-language-action (VLA) models, represents a different branch of research. Within the offline imitation learning framework, diffusion policy stands out as the most effective at modeling multi-modal distributions, which is why we selected it as the backbone for our method.

---

### Official Review · Reviewer_74wV · 2024-11-05

**Soundness:** 2
**Presentation:** 4
**Contribution:** 2
**Rating:** 1
**Confidence:** 4

**Summary:**

This paper proposed a framework that aims to adaptively learn manipulation policy based on previous policy execution results. The framework is based on diffusion policy learning. The manipulation specifically targets articulated object manipulation such as safe and doors. Both simulation and real-world experiments are conducted.

**Strengths:**

- The studied problem in this paper is interesting.
- The paper presentation is good and clear.
- The experiment results look strong and outperform related baselines.

**Weaknesses:**

- There isn't much technical novelty in this paper. It seems that the algorithm/model itself does not guarantee adaptive. The only "adaptive" component stems from adaptive demonstration collection. How is this "adaptive" policy different from vanilla diffusion policies?
- The experiment shows that the proposed method is better than AdaAfford, which is quite surprising. If what the authors of AdaAfford argued is true, AdaAfford should also be able to solve the problem presented in this paper. Why is AdaAfford "not so stable due to the capability limit of cVAE"? Why is the method in this paper stable? How does "stable" defined? Is it because diffusion model vs. cVAE, or it's because output actions vs. point cloud affordance?

**Questions:**

The authors can reply to the questions raised in "Weakness" section.

---

> ### Author Response · Authors · 2024-11-22
> **Response to Reviewer 74wV [Part 1/2]**
>
> Thank you for your review! We would like to address your concerns regarding novelty, adaptive policy and baseline performance in the following response.
>
> **Q1: What is the novelty in this paper?**
>
> We would like to emphasize that adaptive articulated object manipulation, such as opening a bottle or a microwave, represents a set of realistic and frequently encountered tasks for robots. However, the lack of attempts to build the corresponding simulation environment and dataset makes this problem relatively underexplored. Existing simulation environments overlook the complex and realistic adaptive mechanisms inherent in the internal structures of such objects.
>
> Therefore, the primary contribution of our paper lies in introducing the crucial and realistic **adaptive articulated object manipulation task** along with the corresponding novel **AdaManip environment and dataset**. Moreover, we design an **adaptive demonstration collection and policy learning pipeline**, enabling the policy to model multi-modal trajectory distributions and update its posterior based on environmental feedback.
>
> As noted by Reviewer XbhV, our environment "addresses a gap in existing robotic manipulation environments." Similarly, Reviewer eEiz acknowledged the "significant contributions by providing a rich and diverse set of objects, enabling research on the important but under-explored adaptive manipulation problem."
>
> The adaptive articulated object manipulation problem is both **crucial** and **underexplored** in robotic manipulation research. We believe our work addresses the lack of realistic adaptive articulated object manipulation environments and provides an essential resource for the robotics community. Through this work, we aim to pave the way for further research on real-world manipulation challenges and emphasize the importance of collecting adaptive demonstrations over optimal demonstrations for learning adaptive policies.
>
> **Q2: It seems that the algorithm/model itself does not guarantee adaptive. The only "adaptive" component stems from adaptive demonstration collection. How is this "adaptive" policy different from vanilla diffusion policies?**
>
> A policy dealing with adaptive manipulation mechanisms should be equipped with the ability to model multi-modal distributions, which is the strength of diffusion-based imitation learning. However, vanilla diffusion policy fails to ensure adaptiveness, as the policy is typically trained on static optimal success trajectories that do not account for failures during manipulation. We point out in this work that such optimal demo is not adequate for realistic articulated object manipulation policy learning, which is overlooked in existing literatures.
>
> To address this, we developed the adaptive demonstration collection policy that captures recovery from failures, enabling adaptive policy learning. Initially, the policy **prior**, conditioned only on visual observations, predicts multi-modal trajectories. For example, as shown in Figure 1 in our paper, in the open-safe task, the AdaManip policy might predict both directly opening the door and rotating the knob to unlock it. After attempting to pull the handle and failing to open the door, the policy updates its **posterior** based on the feedback, collapsing the multi-modal prediction into a single mode—rotating the knob. By learning from adaptive demonstrations and conditioning on failure feedback, our AdaManip model ensures adaptive manipulation capabilities.
>
> Table 3 demonstrates that policies trained with static optimal demonstrations perform significantly worse than those trained with adaptive demonstrations. Furthermore, Table 4 highlights the importance of avoiding redundant adaptive trials in the demonstrations, validating the effectiveness of our design.

---

> ### Author Response · Authors · 2024-11-22
> **Response to Reviewer 74wV [Part 2/2]**
>
> **Q3: AdaAfford should also be able to solve the problem presented in this paper. Why is AdaAfford not so stable due to the capability limit of cVAE? Why is the method in this paper stable?**
>
> There are **two** main reasons why our method performs better than AdaAfford.
>
> First, diffusion models are more capable of modeling multi-modal distributions, and the adaptive manipulation tasks we focus on naturally require the policy to capture the ambiguity of manipulation trajectories under visual observation. Since our method is based on the more advanced diffusion model compared to the cVAE used by AdaAfford, it is better at learning the adaptive policy.
>
> Second, AdaAfford relies on learning the point-level affordance value distribution, which is trained on both positive and negative action labels obtained from a large number of trials. The point-level affordance space is more complex than the pose action space, and therefore requires more training sample data. However, in our imitation learning setting, we need to train the adaptive policy using limited demonstration trajectories, which is a practical setting adopted by previous methods, including Diffusion Policy [1] and ALOHA [2]. To ensure a fair comparison between the baselines and our method, we use the same quantity of training data. In this case, AdaAfford suffers from inadequate data to model the entire affordance distribution, leading to a significant performance loss.
>
> [1] Chi et al, Diffusion Policy: Visuomotor Policy Learning via Action Diffusion, RSS 2023
>
> [2] Zhao et al, Learning Fine-Grained Bimanual Manipulation with Low-Cost Hardware, RSS 2023

---

> ### Author Response · Authors · 2024-11-29
>
> We sincerely thank you for your thoughtful review. As the discussion period draws to a close, we would appreciate your feedback to confirm whether our replies have addressed your concerns. If you have any remaining questions, we are happy to provide further clarification. If our responses have resolved your concerns, we would be deeply grateful if you could consider raising the score. Thank you again for your time and effort during the review process.

---

> > ### Comment · Reviewer_74wV · 2024-12-02
> >
> > I still have concerns over the novelty of the paper. In my opinion, algorithmically, the so-called "adaptive policy learning" is not that much different than vanilla diffusion policy learning.
> >
> > I also don't believe that the explanation from the authors on the performance difference between the method proposed in this paper vs. AdaAfford makes sense. Even if diffusion policy is better than cVAE, it has nothing to do with the argued novelty in this paper (i.e. "adaptive" policy learning). So I don't understand why such comparison is meaningful for this paper.
> >
> > So I will keep my original rating "3: reject, not good enough".

---

> > > ### Author Response · Authors · 2024-12-02
> > > **Second Round Response to Reviewer 74wV**
> > >
> > > We appreciate your response, although it arrives at the tail end of the discussion period. We would like to address some concerns regarding your review, as we believe there are misunderstandings that hinder a constructive discussion.
> > >
> > > First, we reiterate the main contribution of our paper, which we clearly outlined in our rebuttal: the introduction of the **adaptive articulated object manipulation task**, along with the novel **AdaManip environment and dataset**. We want to emphasize that the introduction of this important and under-explored problem, alongside a new dataset, is a significant contribution to the field—one that goes beyond mere algorithmic improvements. This is exactly why papers like ManiSkill2[1] and DaxBench[2], which focus on introducing new tasks and simulations, are considered influential and have been accepted by ICLR. Additionally, we have carefully designed the adaptive demonstration collection pipeline to ensure adaptive policy learning, a feature that is not guaranteed by vanilla diffusion policies. This aspect is a core part of our contribution, and we have stated this clearly in our response.
> > >
> > > Secondly, we find it unjustified to simply state, "I don’t believe the explanation on the performance difference between AdaManip vs. AdaAfford," without offering any reasoning. If you have specific concerns, we would be more than happy to address them and provide further clarification. We believe that this type of open dialogue is essential for improving the quality of the paper, which is the very purpose of the review process.
> > >
> > > Thirdly, we would like to clarify that the comparison between AdaManip and AdaAfford is both appropriate and meaningful, as AdaAfford also adjusts manipulation policies based on historical data. This makes it a relevant baseline for our work. Moreover, the introduction of adaptive demonstrations is what enables adaptive policy learning in our approach, which is the contribution of our work and we have clearly stated it in our previous response.
> > >
> > > We are eager to engage in a more productive and constructive discussion, and we hope that this response will help clear up any misunderstandings. We look forward to a more objective and thorough evaluation of our work and responses.
> > >
> > > [1] Gu et al, Maniskill2: A unified benchmark for generalizable manipulation skills, ICLR 2023
> > >
> > > [2] Chen et al, Daxbench: Benchmarking deformable object manipulation with differentiable physics, ICLR 2023

---

> > > > ### Comment · Reviewer_74wV · 2024-12-02
> > > >
> > > > Authors:
> > > >
> > > > I appreciate your additional reply. I also apologize for posting my comments late due to a busy schedule on my end.
> > > >
> > > > Let me clarify a few points in your reply.
> > > >
> > > > ---
> > > >
> > > > > First, we reiterate the main contribution of our paper ...
> > > >
> > > > Regarding algorithmic novelty: from your reply, I believe you also agreed that this paper does not contain much algorithmic novelty. I think there isn't much more to debate about.
> > > >
> > > > Regarding the AdaManip environment and dataset: projects such as ManiSkill and DaxBench introduce new robotic simulation frameworks based on new or existing physics engines. On the other hand, I believe your work only creates a few environments with a few new objects based on existing simulation frameworks (e.g. by importing new object meshes, modifying a few lines in the object URDF code etc). The environment and dataset contribution of your work is in no comparison with ManiSkill and DaxBench. It is quite misleading to mention previous ICLR papers such as ManiSkill and DaxBench to argue that your paper should also be accepted by ICLR.
> > > >
> > > > ---
> > > >
> > > > > Secondly, we find it unjustified to simply state, ... Thirdly, ...
> > > >
> > > > If you carefully read my reply, you should realized that I used three sentences to explain why I think your previous explanations make no sense. However, you quoted only my first sentence and then commented "*we find it unjustified to simply state ... without offering any reasoning*". I found such comments really annoying.
> > > >
> > > > Let me explain with a few more sentences just in case you couldn't understand it again: basically, AdaManip and AdaAfford are two algorithms, so the comparison between these two should be on the algorithmic aspect. However, the "cVAE vs. diffusion models" comparison does not justify anything meaningful that is related to the algorithmic novelty of this paper (because you cannot just argue that "*diffusion*" policy is "*adaptive*" policy). On the other hand, the "introduction of adaptive demonstrations" is not on algorithmic but on data aspect, and if it is only fully used by AdaManip but not by AdaAfford, it raises an even worse concern because the comparison is not fair in terms of how the "*adaptive*" data is used.
> > > >
> > > > This is why I don't believe that the explanation from the authors on the performance difference between AdaManip vs. AdaAfford makes sense.
> > > >
> > > > ---
> > > >
> > > > I still hold my view of voting for rejection of this paper, due to the reasons stated above.

---

> ### Author Response · Authors · 2024-12-02
>
> Thank you for your further detailed reply! We greatly appreciate the time and effort you’ve taken to explain your concerns. We would like to take this opportunity to address some misunderstandings and clarify our work further.
>
> First, when we refer to ManiSkill2 and DaxBench, our intention is not to directly compare our work with these works, but rather to emphasize that the dataset/simulation environment itself is a significant contribution that goes beyond algorithmic innovation. As we stated in our response, the adaptive manipulation problem is important yet under-explored, and our dataset aims to help close this gap. The data synthesis and simulation mechanisms were developed with considerable effort, as the objects were not simply borrowed from existing datasets but were manually crafted and annotated. Additionally, the environments themselves required significant coding to create, further emphasizing the level of effort involved. Our dataset addresses the lack of realistic adaptive articulated object manipulation environments, providing an essential resource for the robotics community that will support future research into real-world manipulation challenges. We believe this constitutes a meaningful contribution to the field.
>
> Secondly, thank you once again for your detailed response. We now have a clearer understanding of the concerns you raised, and we appreciate the opportunity to clarify. You mention that "the introduction of adaptive demonstrations is not on algorithmic but on data aspect, and if it is only fully used by AdaManip and not by AdaAfford, it raises a concern because the comparison is not fair in terms of how the adaptive data is used." However, we want to clarify that AdaAfford is trained on the **same data** as AdaManip—the adaptive demonstrations collected by our designed policy. Therefore, we believe the comparison is indeed fair, as both methods use the same adaptive data for training.
>
> The reason why we find the criticism in your previous response somewhat unclear is that we had difficulty understanding the connection between the statement "I don’t believe the explanation" and your subsequent comment, "Even if diffusion policy is better than cVAE, it has nothing to do with the argued novelty in this paper (i.e. 'adaptive' policy learning). So I don’t understand why such comparison is meaningful for this paper." In your initial review, you raised a concern about why the performance of AdaManip is better than AdaAfford, and we provided an explanation highlighting the differences between diffusion and cVAE policies, as well as the point-level training nature of AdaAfford, which requires additional training data. We believed that this explanation addressed your concerns about the performance gap. Therefore, we were uncertain about the reasoning behind the dismissal of our explanation, and we appreciate your further clarification that prompts us to explain that the data used by both methods are the same.
>
> Regarding the meaning of the comparison: since both models are trained on the same data, we feel that comparing AdaManip and AdaAfford is valuable, as it highlights how different design choices in the model architecture and training pipeline can better support the training of adaptive policies for the tasks we propose. As noted in our first round response and recognized in your reply, diffusion policies alone do not ensure adaptiveness, which underscores the importance of adaptive demonstrations—one of the core contributions of our work. The ablation experiments presented in Table 3 are intended to evaluate the data aspect, and the comparison with AdaAfford helps illustrate the effectiveness of diffusion models and history-conditioned action imitation, as opposed to affordance learning.
>
> We sincerely hope our response has addressed your concerns, and if there are any additional points you would like to discuss, we would be more than happy to respond to them. Once again, we truly appreciate the time you’ve taken to elaborate on your feedback and look forward to any further suggestions you may have.

---

### Official Review · Reviewer_ooAE · 2024-11-05

**Soundness:** 3
**Presentation:** 3
**Contribution:** 3
**Rating:** 6
**Confidence:** 4

**Summary:**

In this work, the authors investigate the task of learning manipulation policies for articulated objects. Specifically, they recognize a gap in the literature relating to the intersection between articulation and locking/sequential mechanisms that are often required in real-world articulated object manipulation scenarios (e.g. unlock the door before opening). Their contributions are roughly twofold: first, they design a set of novel articulated object simulation environments that model the locking/sequential mechanisms during articulation tasks; second, they propose a demonstration dataset and policy architecture to learn novel policies to tackle these environments. They demonstrate the effectiveness of their policies on their simulated environments and real-world embodiments of the same.

**Strengths:**

# Originality

While locking mechanisms and articulated objects have been studied (and modeled) independently, the interface of the two has not yet been particularly well-studied, despite the fact that many articulated objects require navigating locking/latching mechanisms to accomplish opening. Proposing tasks that model this aspect of the task is a good original contribution.

On the policy side, I can’t fully evaluate whether the design of the policy contains much novelty/originality, as it appears to be a straightforward modification of diffusion policies, but without enough architectural detail provided to assess novelty.

# Quality

The environment-modeling side of the paper seems high quality, with good diversity over tasks and attention to relevant details for releasing a new benchmark. The proposed policy and its analysis is of OK quality - not enough detail is provided in the experiments section for me to evaluate.

# Clarity

The paper is clear and well-written (except figures, which I will discuss later).


# Significance

The paper is of moderate significance, as this set of tasks (and an environment which accurately models it) is indeed important to study and is currently underexplored. Difficult to assess the significance of the policy proposal, however.

**Weaknesses:**

There are 3 primary categories of weakness
* The policy itself was not fully characterized. Not many details were provided as to the architecture, training procedure, etc. and I was not able to really assess in comparison to existing literature. In particular, the treatment of point clouds and history were glossed over, when in other works it has been shown that these details matter for different kinds of generalization. Additionally, the ablations don’t provide much insight into why the design decisions of the policy lead to improved performance over the baselines - “Ours” without history is much stronger than the baselines.
* Experimental details: It was unclear whether the baselines were fairly conducted - were the VAT-MART and AdaAfford methods trained on the same data as “Ours”? Or were pretrained weights used?
* Generalization characterization: when releasing benchmarks with metrics, ideally we would also receive a set of metrics with a clear experimental protocol, as well as what conditions/generalization criteria the experiment is trying to measure. Overall “success rate” for a single kind of evaluation is somewhat coarse. Instead, it would be desirable to have a clear set of different benchmarks, e.g. to measure cross-instance generalization, generalization across initialization/viewpoint, cross-class generalization, etc.



Smaller weaknesses
* Figure 1 is too dense, and the symbols unintuitive. More text should be incorporated into the diagram, it’s really hard to figure out what’s going on.

**Questions:**

* Why wasn’t 3D Diffusion policy considered as a policy class? https://3d-diffusion-policy.github.io/
* Why was the decision made to use a logical lock/mechanism for task progression instead of a physical mechanism?
* How is success rate defined?
* How many history steps are needed for success across the tasks? Would be good to understand this mechanism a bit more.
* How is sim2real accomplished? Is a new dataset collected in real setting? Or is it transferred zero-shot.

# Update post-rebuttal

The authors have done a good job responding to my questions. In particular, it is good to see that the DP3 baseline and their policy baseline are roughly equivalent in performance, matching my expectations. It's also good to see that additional high-quality recovery demonstrations improve success rates. This makes it a stronger paper - although because the policy-level changes are primarily a data contribution rather than a policy contribution, I maintain that my current score of 6 (marginal accept) is warranted.

---

> ### Author Response · Authors · 2024-11-22
> **Response to Reviewer ooAE [Part 1/3]**
>
> Thank you for your valuable comments! We hope to address your concerns in the following response.
>
> **Q1: More details in the policy. The policy itself was not fully characterized. Not many details were provided as to the architecture, training procedure, etc.**
>
> AdaManip policy models the conditional distribution $P(A_t|O_t,\hat{A}_t)$. Here $A_t$ refers to the predicted action sequence, $O_t$ refers to the observation history, including 3D point clouds and proprioception states, and $\hat{A}_t$ refers to action history.
>
> The AdaManip policy is composed of two models: a **PointNet++ vision encoder** and a **U-Net-based noise prediction network**. The PointNet++ encoder processes 3D point clouds into latent embeddings. These embeddings are concatenated with proprioception states and previous actions to form a comprehensive history condition vector, which is then used as input to the noise prediction network. The noise prediction network generates the noise conditioning on the history and denoising iteration k, for the denoising scheduler, ultimately producing the predicted action sequence.
>
> The training procedure is one-stage. After collecting the demonstration data, it is processed into pairs of $(O_t,\hat{A}_t| A_t)$ to enable end-to-end training of both the PointNet++ encoder and the noise prediction network. The training follows the loss function specified in Equation 2 of Section 4.2:
>
> $L(\theta) = E_{k \sim \mathcal{U}(0, 100), \epsilon^{k} \sim \mathcal{N}(0, \sigma_{k}^2 I)}\left[ \left\Vert \epsilon^{k} - \epsilon_{\theta}(A_t^0+\epsilon^k, O_t, \hat{A}_t, k) \right\Vert_2^2  \right]$
>
> If you have further concerns regarding the policy architecture or training procedure, we would be happy to provide additional details and explanations.
>
> **Q2: The ablations don’t provide much insight into why the design decisions of the policy lead to improved performance over the baselines - “Ours” without history is much stronger than the baselines.**
>
> The ablation method is ours without adaptive demonstration data, maintaining the diffusion-based imitation learning. There are two main reasons why this ablation performs much better than the affordance-based baselines.
>
> First, diffusion models are inherently more capable of modeling multi-modal distributions. The adaptive manipulation tasks we focus on naturally involve ambiguities in manipulation trajectories under visual observation. Although the ablation method is trained solely on static success trajectories and cannot efficiently recover from failures, it can still model the multiple possible successful manipulation trajectories of the mechanism. As a result, it can possibly sample corresponding trajectories based on the current object state. In contrast, affordance-based baselines use cVAE to model multi-modal distributions, which is less effective compared to diffusion models.
>
> Second, affordance-based methods learn the point-level affordance value distribution, which is trained on both positive and negative action labels obtained from a large number of trials. The point-level affordance space is more complex than the pose action space, and therefore requires more training sample data. However, in our imitation learning setting, we need to train the adaptive policy using limited demonstration trajectories, which is a practical setting adopted by previous methods, including Diffusion Policy [1] and ALOHA [2]. To ensure a fair comparison between the baselines and our method, we use the same quantity of training data. In this case, the baselines suffer from inadequate data to model the entire affordance distribution, leading to a significant performance loss.
>
> **Q3: It was unclear whether the baselines were fairly conducted - were the VAT-MART and AdaAfford methods trained on the same data as “Ours”? Or were pretrained weights used?**
>
> The VAT-MART and AdaAfford were trained on the same data as our method, without using any pretrained weights. This approach ensures a fair comparison between the methods.

---

> ### Author Response · Authors · 2024-11-22
> **Response to Reviewer ooAE [Part 2/3]**
>
> **Q4: Generalization characterization and experimental protocol. How is success rate defined? It would be desirable to have a clear set of different benchmarks, e.g. to measure cross-instance generalization, generalization across initialization/viewpoint, cross-class generalization, etc.**
>
> The testing is conducted on the training shapes at the category level, with variations in the object positions, poses (orientations), and joint configurations in both simulation and real-world experiments. Such setting aligns with current imitation learning studies [1, 2] using few-shot demonstrations (in our work, we use only 20 demos per shape for training).
>
> The success rate is defined as the total number of successful manipulation trials divided by the total number of objects $N$ times the number of test episodes $T$ for each object: $SR = \frac{\sum_i\sum_j I_{ij}}{N * T}$, where $i$,$j$ are indices for the object and test episode, respectively.
>
> As for the generalization towards novel shapes and categories, another branch of current works is trying to use large-scale data [3] or foundation models [4] to achieve such generalization capability, which is orthogonal to the contribution to our work: empowering imitation learning with the adaptive manipulation capability on diverse object mechanisms with our proposed environment, mechanisms and assets.
>
> We agree that incorporating different benchmark settings would enhance the comprehensiveness of the AdaManip environment and dataset. In the release of the AdaManip dataset, we promise to provide a clear set of different benchmarks including cross-instance/class/initialization/viewpoint generalization. It is a promising direction for the following work to further study how to leverage the generalizable knowledge inherited from large models to enhance the cross-shape and cross-category generalization capability of adaptive manipulation on diverse object mechanisms.
>
> Thank you for the valuable comments!
>
> **Q5: Why wasn’t 3D Diffusion policy considered as a policy class?**
>
> Thank you for the suggestion! We conducted additional experiments to evaluate 3D Diffusion Policy as a baseline. The quantitative results in the table below demonstrate that 3D Diffusion Policy outperforms affordance-based methods, as the diffusion model is more effective at modeling multi-modal distributions with fewer demonstrations. However, it performs worse than AdaManip due to the absence of the adaptive demonstration collection pipeline and action history conditioning.
>
>
> | **Method**    | **Bottle**          | **Pen**             | **PressureCooker** | **CoffeeMachine**  | **Safe**            | **Microwave**       | **Window**         | **Door**            | **Lamp**            |
> | --- | --------------- | --------------- | -------------- | --------------- | --------------- | --------------- | -------------- | --------------- | --------------- |
> | DP3    | 83.16$\pm$12.19 | 83.00$\pm$11.87 | 86.67$\pm$4.08 | 85.71$\pm$10.10 | 35.71$\pm$21.02 | 62.00$\pm$13.27 | 70.95$\pm$3.37 | 58.75$\pm$14.84 | 53.57$\pm$13.20 |
>
> **Q6: Why was the decision made to use a logical lock/mechanism for task progression instead of a physical mechanism?**
>
> The implementation of mechanism simulation combines **high-level discrete conditions** and **physical simulation**. We define the lock state as a discrete condition that updates the joint upper limit, which subsequently affects the physical simulation. As described in Section 5.3, we track the state of the key part’s joint to determine the lock state. When the lock state transitions to "unlock," the part joint limit is lifted, allowing the part to be opened.
>
> Taking the case of a bottle for example, the cap must be rotated to a certain angle before it can be lifted. In the simulator, we track the cap's revolute joint angle, and when it exceeds the threshold, we set the `lock_state` to `False` and increase the prismatic joint's upper limit from 0 to a non-zero value. This change allows the cap to be lifted in the physical simulation.
>
> We choose not to simulate complex lock collision shapes, as such simulations could introduce uncontrollable errors such as interpenetration (more details discussed in [5], Appendix C), leading to instability. Since our focus is on simulating realistic adaptive manipulation mechanisms, updating the joint limits based on discrete lock states provides sufficient realism while maintaining simulation stability. Therefore, we opted for the current implementation rather than using complex shape collision simulations.

---

> ### Author Response · Authors · 2024-11-22
> **Response to Reviewer ooAE [Part 3/3]**
>
> **Q7: How many history steps are needed for success across the tasks?**
>
> For the tasks in this paper, a history length of 4 is sufficient for the diffusion-based model to capture the necessary failure information and update the policy distribution effectively.
>
> This relatively short context works because we employ **trajectory sparsification**, which removes intermediate states and retains only keyframes. If the history consists of a dense trajectory of the robot's end-effector poses, the policy would require a longer history context to capture previous failures. However, training a policy with a long history context is computationally expensive and less robust. To address this, we sparsify the trajectory by saving only the keyframes of the demonstration trajectories for imitation learning.
>
> For example, in the open-safe task (the most complex task among all), the recorded history includes grasping poses and several manipulation poses while omitting most intermediate steps. The history condition for the policy is as follows: [grasp the handle, pull the door and fail to open]. Other intermediate poses during execution are excluded from the history. Based on this context, the robot predicts the next goal pose [unlock the key]. Once the goal pose is predicted, we apply inverse kinematics (IK) to plan the path for execution.
>
> By abstracting the demonstration trajectories into keyframes, an observation history length of 4 is sufficient to capture all relevant information, ensuring computational efficiency and robust policy performance.
>
> **Q8: How is sim2real accomplished? Is a new dataset collected in real setting? Or is it transferred zero-shot.**
>
> We collect real-world demonstrations via human teleoperation to train the policy for real-world settings, and thus there is no sim-to-real gap.
>
> **Q9: Figure 1 is too dense, and the symbols unintuitive. More text should be incorporated into the diagram, it’s really hard to figure out what’s going on.**
>
> Thank you for the valuable suggestion! We agree that incorporating more text into Figure 1 could improve clarity and make the motivation easier to understand. We will revise the figure accordingly to better convey the key ideas and ensure it is more intuitive. Thank you again for your helpful feedback!
>
> [1] Chi et al, Diffusion Policy: Visuomotor Policy Learning via Action Diffusion, RSS 2023
>
> [2] Zhao et al, Learning Fine-Grained Bimanual Manipulation with Low-Cost Hardware, RSS 2023
>
> [3] Yu et al, RT-1: Robotics Transformer for real-world control at scale, ArXiv 2022
>
> [4] Yan et al, DNAct: Diffusion Guided Multi-Task 3D Policy Learning, ArXiv, 2024
>
> [5] Tang et al, Automate: Specialist and generalist assembly policies over diverse geometries, RSS 2024

---

### Official Review · Reviewer_14u1 · 2024-11-07

**Soundness:** 4
**Presentation:** 1
**Contribution:** 3
**Rating:** 5
**Confidence:** 5

**Summary:**

The authors present an imitation learning study focusing on tasks involving articulated objects. The study necessitates (1) a multi-modal action proposal and (2) adaptive manipulation based on historical actions.

The author conducted experiments both in simulation and real-world.

**Strengths:**

The motivation of the paper is sound. It is an important problem in imitation learning: (1) multi-modal action proposal and (2) adaptive manipulation from history actions.

**Weaknesses:**

Could the authors clarify any generalizations observed in the simulation results? Further details on the differences between the training and testing datasets would be helpful.

The manuscript requires significant revisions, particularly Figure 1 and 3, which is quite confusing. If I understand correctly, you employed a rule-based policy described as "adaptive" to collect data. However, it appears that this policy does not incorporate any prior knowledge about the object’s mechanism for the learning phase. Could you clarify this?

The real-world results presented are not convincing. There is a lack of discussion on generalization, and the evaluation seems overly simplistic with limited data. The methods used for collecting demonstrations in the real world are also unclear. Additionally, it is not specified whether the data collection method used in simulations is scalable to real-world applications.

I would like to see more discussion on history information part. How did you incorporate history? would simply adding history cause overfitting problem?

**Questions:**

Why did you choose point cloud input rather than RGB?

Further details about the real-world experiments are needed, specifically:

1. What type of low-level controller was used?
2. To what extent does the training generalize to the test scenarios?
3. What is the frequency of the policy?
4. How was the expert demonstration collected?
-- Understanding the data collection process is crucial. Are 35 demonstrations sufficient to capture the task's multi-modal nature?

---

> ### Author Response · Authors · 2024-11-22
> **Response to Reviewer 14u1 [Part 1/2]**
>
> Thank you for the insightful comments! We will provide our point-by-point response and hope our response helps address your concerns.
>
> **Q1: The generalization ability of the method.**
>
> The testing is conducted on the training shapes at the category level, with variations in the object positions, poses (orientations), and joint configurations in both simulation and real-world experiments. Such setting aligns with current imitation learning studies [1, 2] using few-shot demonstrations (in our work, we use only 20 demos per shape for training).
>
> As for the generalization towards novel shapes, another branch of current works is trying to use large-scale data [3] or foundation models [4] to achieve such generalization capability, which is orthogonal to the contribution to our work: empowering imitation learning with the adaptive manipulation capability on diverse object mechanisms with our proposed environment, mechanisms and assets.
>
> It is a promising direction for the following work to further study how to leverage the generalizable knowledge inherited from large models to enhance the cross-shape and cross-category generalization capability of adaptive manipulation on diverse object mechanisms.
>
>
> **Q2: Figure 1 & 3 are confusing. A rule-based policy described as "adaptive" is employed to collect data. However, it appears that this policy does not incorporate any prior knowledge about the object’s mechanism for the learning phase. Could you clarify this?**
>
> This rule-based demo collection policy is designed to cover all the possible failure and success states (i.e., all possible manipulation rules) during the object manipulation, as well as the best recovery actions conditioning on failure.
>
> For example, in the open-safe task, the demonstration policy follows these steps: 1. The robot first pulls the door handle. If successful, it proceeds to open the door. 2. If pulling the handle fails, the robot switches to rotating the knob. 3. It randomly chooses a rotation direction and switches to the other direction if the first attempt fails. 4. Once the knob is successfully rotated to unlock the door, the robot pulls the handle again to complete the task.
>
> By training on the data generated by the rule-based demo collection policy, the AdaManip policy captures the multi-modal prior distribution and develops the ability to adjust its posterior based on feedback. For instance, at the start of the open-safe task, the policy predicts multiple possibilities, such as directly opening the door or rotating the knob to unlock it. If the robot attempts to pull the handle and fails, the policy updates its posterior based on the feedback, collapsing the multi-modal prediction into a single mode—rotating the knob.
>
> Thank you for your suggestion regarding the figures! We agree that certain modifications and clarification would improve their clarity and will update them accordingly in the revised paper.
>
> [1] Chi et al, Diffusion Policy: Visuomotor Policy Learning via Action Diffusion, RSS 2023
>
> [2] Zhao et al, Learning Fine-Grained Bimanual Manipulation with Low-Cost Hardware, RSS 2023
>
> [3] Yu et al, RT-1: Robotics Transformer for real-world control at scale, ArXiv 2022
>
> [4] Yan et al, DNAct: Diffusion Guided Multi-Task 3D Policy Learning, ArXiv, 2024

---

> ### Author Response · Authors · 2024-11-22
> **Response to Reviewer 14u1 [Part 2/2]**
>
> **Q3: How was the expert demonstration collected? Are 35 demonstrations sufficient to capture the task's multi-modal nature? Is the data collection method used in simulations scalable to real-world applications?**
>
> In the real-world experiments, we collect 35 adaptive expert demonstrations for each object by human teleoperation. The Franka Emika Panda Robot Arm is reset to the initial pose with randomization at the beginning of each trial.
>
> 35 demonstrations are sufficient to capture the multi-modal nature. For example, in the open-safe task, which is the most complex among all nine categories, the macro action set is {grasp the door handle, grasp the knob, rotate the knob clockwise, rotate the knob counterclockwise, pull the door}. All trajectories, including both direct successes and recoveries from failures, will not exceed 10 distinct possibilities. Therefore, 35 trajectories are adequate to cover all modalities.
>
> In the simulation, we employed rule-based methods for data collection, which rely on object part pose annotations and human-designed policies. These methods are scalable to real-world settings, as similar policies can be designed and poses annotated manually. However, as the complexity and diversity of mechanisms increase, designing pipelines for each category may become impractical. A promising direction for future work is to explore methods to automate or streamline the data collection process.
>
> **Q4: More discussion on history information part. How did you incorporate history? would simply adding history cause overfitting problem?**
>
> The observation history includes 3D point clouds and proprioception states. The 3D point clouds are processed by a PointNet++ model and transformed into latent embeddings. These point cloud embeddings are then concatenated with proprioception states and previous actions to form a comprehensive history condition vector, which serves as the input to the noise prediction network of the diffusion model.
>
> Adding history does not cause overfitting; rather, it helps prevent the model from fitting to certain static trajectories: The diffusion model trained on adaptive demonstrations learns to adjust its posterior dynamically based on observation and action history.
>
> For example, in the open-safe task, the policy might initially predict multiple possibilities, such as directly opening the door or rotating the knob to unlock it. If the robot attempts to pull the handle and fails to open the door, the policy updates its posterior based on the feedback, collapsing the multi-modal prediction into a single mode—rotating the knob. By conditioning on failure history, the AdaManip model avoids overfitting particular static trajectories.
>
> **Q5: Why did you choose point cloud input rather than RGB?**
>
> RGB images lack depth information, which is crucial for accurate object pose estimation. Adaptive manipulation tasks require predicting action poses in 3D space, which necessitates 3D information. Point cloud inputs provide this essential depth information, making them more suitable for our approach.
>
> **Q6: What type of low-level controller was used in the real-world experiment?**
>
> We use catesian impedance controller for the Franka Emika Panda Robot Arm in real-world experiments.
>
> **Q7: What is the frequency of the policy?** The frequency is about 50Hz.

---

> ### Author Response · Authors · 2024-11-29
>
> We sincerely thank you for your thoughtful review. As the discussion period draws to a close, we would appreciate your feedback to confirm whether our replies have addressed your concerns. If you have any remaining questions, we are happy to provide further clarification. If our responses have resolved your concerns, we would be deeply grateful if you could consider raising the score. Thank you again for your time and effort during the review process.

---

### Official Review · Reviewer_eEiz · 2024-11-09

**Soundness:** 3
**Presentation:** 3
**Contribution:** 3
**Rating:** 8
**Confidence:** 4

**Summary:**

This paper introduces AdaManip, a new environment and method for training robots to manipulate articulated objects with adaptive mechanisms.  Existing datasets often simplify these interactions, allowing objects to be manipulated without accounting for hidden states like locked latches or required rotation angles.  AdaManip addresses this by incorporating five key adaptive mechanisms: lock mechanisms, random rotation direction, rotate & slide, push/rotate, and switch contact.  The environment includes nine object categories (bottles, pens, coffee makers, windows, pressure cookers, lamps, doors, safes, and microwaves) with diverse instances, totaling 277 objects.

The authors also propose an adaptive demonstration collection pipeline that uses rule-based expert policies to generate trajectories reflecting realistic partial observation scenarios. These demonstrations are then used to train a 3D visual diffusion-based imitation learning model.

Experiments in simulation and real-world settings with a Franka Panda robot demonstrate the effectiveness of AdaManip. The proposed method outperforms baseline approaches like VAT-Mart and AdaAfford, achieving higher success rates across all object categories.  Ablation studies confirm the importance of adaptive demonstrations and the strategy of limiting each demonstration to a single adaptive trial. The real-world experiments further validate the policy's ability to generalize and adapt to real-world objects and conditions.

**Strengths:**

- This paper points out an important but under-explored problem: the adaptive manipulation of articulated objects with complex mechanisms. Previous work primarily focused on simpler manipulations, neglecting the realistic challenges of hidden object states.
- The AdaManip environment and associated dataset are significant contributions. They provide a rich and diverse set of objects, as well as the rule-based demonstrations, enabling research on more realistic manipulation tasks.
- The categorization of five key adaptive mechanisms (lock mechanisms, random rotation direction, rotate & slide, push/rotate, and switch contact) provides valuable insights and serves as an inspiring framework for creating new tasks.
- The application of 3D visual diffusion-based imitation learning is a good baseline for this problem.  Diffusion models are well-suited to capturing the multi-modal nature of the task, where multiple valid manipulation trajectories can exist for a given observation.
- The real-world experiments are valuable, demonstrating the practicality of the approach.
- The paper is generally well-written and easy to follow.

**Weaknesses:**

- Relying on rule-based expert policies for demonstration collection can limit the complexity of the learned behaviors. Exploring methods for learning these demonstrations or incorporating human demonstrations could lead to more robust and generalizable policies.

- The source code and dataset have not been released.

- Environmental specifications, including frame rate and API details, are not provided.

- The paper primarily compares against affordance-based methods.  A comparison to other offline imitation learning methods (e.g., [1]) and online learning-from-demo methods (e.g., [2][3][4]) would provide a more complete picture of the method's performance.

[1] Zhao, Tony Z., et al. "Learning fine-grained bimanual manipulation with low-cost hardware." arXiv preprint arXiv:2304.13705 (2023).

[2] Ball, Philip J., et al. "Efficient online reinforcement learning with offline data." International Conference on Machine Learning. PMLR, 2023.

[3] Rajeswaran, Aravind, et al. "Learning complex dexterous manipulation with deep reinforcement learning and demonstrations." arXiv preprint arXiv:1709.10087 (2017).

[4] Ho, Jonathan, and Stefano Ermon. "Generative adversarial imitation learning." Advances in neural information processing systems 29 (2016).

**Questions:**

- How is the "unlock mechanism" and other similar mechanisms simulated? Is this based on a specific collision shape or on some high-level discrete conditions?
- How is the real-world demonstration collected?
- Could you provide the detailed hyperparameter values for the diffusion policy, such as the action horizon and prediction horizon?

---

> ### Author Response · Authors · 2024-11-22
> **Response to Reviewer eEiz [Part 1/2]**
>
> We sincerely appreciate your thoughtful feedback and acknowledgment of our work’s significance. Below, we address your concerns in detail.
>
> **Q1: Relying on rule-based expert policies for demonstration collection can limit the complexity of the learned behaviors. Exploring methods for learning these demonstrations or incorporating human demonstrations could lead to more robust and generalizable policies.**
>
> Currently, we rely on rule-based expert policies in the simulator. However, for real-world experiments, expert demonstrations are collected through human teleoperation.
>
> Exploring learning-based methods to generate more diverse demonstrations is indeed an interesting direction, as it could enhance the robustness and generalizability of the learned policies. One potential approach is leveraging pretrained large visual-language models to generate manipulation trajectories. Adding random noise to the generated action sequences could simulate the jitter present in human manipulation, further improving the robustness of the policy.
>
> **Q2: Will the source code be released? Environmental specifications, including frame rate and API details, are not provided.**
>
> We will release the code in an anonymous repository soon. The environment operates at a frame rate of 60, and additional API details will be included in the accompanying documentation. We are committed to maintaining the project to ensure it is user-friendly and easy to follow.
>
> **Q3: The paper primarily compares against affordance-based methods. A comparison to other offline imitation learning methods (e.g., [1]) and online learning-from-demo methods (e.g., [2][3][4]) would provide a more complete picture of the method's performance.**
>
> Thank you for your suggestion! In our imitation learning setting, the adaptive policy is trained using a limited number of demonstration trajectories, which aligns with the practical setting adopted by previous methods, such as Diffusion Policy and ALOHA. Therefore, it would not be a fair comparison to evaluate against RL methods that rely on online data collection beyond offline demonstrations.
>
> To address your feedback, we conducted additional experiments evaluating ACT (ALOHA) as another offline imitation learning baseline. The quantitative results are shown in the following table. While ACT outperforms affordance-based methods due to its ability to model distributions with fewer demonstrations, it performs worse than AdaManip. This is primarily because ACT lacks both the adaptive demonstration collection pipeline and the strong multi-modality modeling capability provided by diffusion models.
>
> | **Method**    | **Bottle**         | **Pen**             | **PressureCooker**  | **CoffeeMachine**  | **Safe**            | **Microwave**       | **Window**          | **Door**           | **Lamp**            |
> | --- | -------------- | --------------- | --------------- | --------------- | --------------- | --------------- | --------------- | -------------- | --------------- |
> | ACT   | 75.79$\pm$8.55 | 74.00$\pm$16.85 | 81.11$\pm$5.49 | 90.48$\pm$6.73 | 28.57$\pm$15.43 | 59.00$\pm$14.46 | 66.67$\pm$5.39 | 52.50$\pm$14.58 | 51.19$\pm$13.62 |

---

> ### Author Response · Authors · 2024-11-22
> **Response to Reviewer eEiz [Part 2/2]**
>
> **Q4: How is the "unlock mechanism" and other similar mechanisms simulated? Is this based on a specific collision shape or on some high-level discrete conditions?**
>
> The implementation combines **high-level discrete conditions** and **physical simulation**. We define the lock state as a discrete condition that updates the joint upper limit, which subsequently affects the physical simulation. As described in Section 5.3, we track the state of the key part’s joint to determine the lock state. When the lock state transitions to "unlock," the part joint limit is lifted, allowing the part to be opened.
>
> For example, consider the case of a bottle. The cap must be rotated to a certain angle before it can be lifted. In the simulator, we track the cap's revolute joint angle, and when it exceeds the threshold, we set the `lock_state` to `False` and increase the prismatic joint's upper limit from 0 to a non-zero value. This change allows the cap to be lifted in the physical simulation.
>
> We choose not to simulate complex lock collision shapes, as such simulations could introduce uncontrollable errors such as interpenetration (more details discussed in [1], Appendix C), leading to instability. Since our focus is on simulating realistic adaptive manipulation mechanisms, updating the joint limits based on discrete lock states provides sufficient realism while maintaining simulation stability. Therefore, we opted for the current implementation rather than using complex shape collision simulations.
>
> Similarly, for other mechanisms, the implementation also relies on a combination of high-level discrete conditions and physical simulation. For instance, in the Random Rotation Direction mechanism, we randomly sample the rotation direction (clockwise or counterclockwise) as a discrete condition. This condition is used to adjust the revolute joint limits, which then influence the physical simulation.
>
> **Q5: How is the real-world demonstration collected?**
>
> We collect 35 adaptive expert demonstrations for each object by human teleoperation. The Franka Emika Panda Robot Arm is reset to the initial pose with randomization at the beginning of each trial. During inference time, our policy takes the observed cropped point cloud and robot arm joint state and outputs the end-effector keyframe action.
>
> **Q6: Could you provide the detailed hyperparameter values for the diffusion policy, such as the action horizon and prediction horizon?**
>
> In our experiments, the prediction horizon is 4, and the action horizon is 2. We will include the detailed hyperparameters of the diffusion model in the appendix of the revised paper. Thank you for the helpful suggestion!
>
> [1] Tang et al, Automate: Specialist and generalist assembly policies over diverse geometries RSS 2024

---

> > ### Comment · Reviewer_eEiz · 2024-11-25
> >
> > Thank the authors for the detailed response!
> >
> > > To address your feedback, we conducted additional experiments evaluating ACT (ALOHA) as another offline imitation learning baseline.
> >
> > According to your results, ACT is a stronger baseline than existing baselines. I suggest including these results in your manuscript.
> >
> > > In our experiments, the prediction horizon is 4, and the action horizon is 2.
> >
> > It seems the prediction horizon and action horizon are quite short. Any intuition behind this hyperparameter selection?

---

> > > ### Author Response · Authors · 2024-11-26
> > >
> > > Thank you for the response and valuable suggestion! We will include the ACT results in the revised paper as recommended.
> > >
> > > The relatively short horizon is effective for our model because we employ **trajectory sparsification**, which removes intermediate states and retains only key frames. If the demonstrations include dense trajectories of the robot's end-effector poses, the policy would require a longer context to capture previous failures. However, training a policy with a long context is computationally expensive and less robust. To address this, we sparsify the trajectory by saving only the key frames of the demonstration for imitation learning.
> > >
> > > For example, in the open-safe task, the recorded trajectories include grasping poses and several manipulation poses while omitting most intermediate steps. The history condition for the policy is as follows: [grasp the handle, pull the door and fail to open]. Other intermediate poses during execution are excluded. Based on this context, the robot predicts the next sub-goal poses [grasp the key, rotate the key]. Once the sub-goal pose is predicted, we apply inverse kinematics (IK) to plan the path for execution.
> > >
> > > By abstracting the demonstration trajectories into keyframes, a horizon length of no more than 4 is sufficient to capture all relevant information and predict adaptive actions, ensuring computational efficiency and robust policy performance.

---

> > > > ### Comment · Reviewer_eEiz · 2024-11-27
> > > >
> > > > Thank you for your response. I don't have further questions.

---

### Author Response · Authors · 2024-11-29
**Revision Summary**

We thank all the reviewers for their comments and suggestions! We have thoroughly revised and updated our paper accordingly, and we summarize the key changes below:

+ Additional evaluations of three baselines will be conducted to highlight the contributions of our method better
+ Enhance Figures 1 and 3 to improve clarity
+ Clarify that the object assets in our AdaManip environment are newly generated, rather than directly borrowed from PartNet or other existing datasets
+ More details on the policy architecture and action space in Section 4.1 and Appendix C
+ Clarify that the real-world demonstrations are collected by human teleoperation

We hope these revisions, along with our responses, address the reviewers' concerns and better showcase the contributions of our work.

---

### Meta-Review · Area_Chair_UJJX · 2024-12-24

**Metareview:**

Summary

This paper introduces a benchmark for articulated object manipulation, addressing the limitations of existing datasets that do not adequately capture the sequential nature of such tasks. The proposed benchmark explicitly models this sequential aspect and evaluates existing policy networks within this framework. Furthermore, the authors extend an existing policy architecture (diffusion policy) by incorporating a mechanism to adapt behavior based on past actions.

Strengths

- The paper identifies a significant gap in current benchmarks related to adaptive object manipulation.
- Based on these findings, the authors have developed a benchmark and simulation environment for the broader community to advance the field.
- The paper evaluates baseline models and proposes strategies for building effective models using the benchmark.

Weaknesses

- The paper offers limited algorithmic novelty, which is acceptable given its primary focus on benchmark development.
- As noted by reviewers, the reliance on rule-based expert policies for demonstration collection may restrict the complexity of the learned behaviors.

The paper received mixed reviews, with three reviewers recommending acceptance and three recommending rejection. The work was extensively discussed during the rebuttal period. The AC acknowledges the reviewers' concerns regarding the lack of significant algorithmic contributions. However, the AC believes the benchmark's contribution outweighs these concerns. The benchmark has substantial potential to advance research in adaptive manipulation. Consequently, the AC recommends accepting this paper.

**Additional Comments On Reviewer Discussion:**

This paper received mixed opinions and was extensively discussed during the rebuttal phase. As highlighted earlier, the benchmark itself holds significant value for the research community. The AC also acknowledges that not all papers need to make algorithmic contributions to be impactful. Therefore, the AC recommends accepting the paper.

---

### Decision · Program_Chairs · 2025-01-22

Accept (Poster)